# Benchmarking Robustness to Adversarial Image Obfuscations

**Florian Stimberg**
Google DeepMind

**Ayan Chakrabarti**
Google Research

**Chun-Ta Lu**
Google Research

**Hussein Hazimeh**
Google Research

**Otilia Stretcu**
Google Research

**Wei Qiao**
Google Ads Safety

**Yintao Liu**
Google Ads Safety

**Merve Kaya**
Google Research

**Cyrus Rashtchian**
Google Research

**Ariel Fuxman**
Google Research

**Mehmet Tek**
Google Ads Safety

**Sven Gowal**
Google DeepMind

## Abstract

Automated content filtering and moderation is an important tool that allows online platforms to build striving user communities that facilitate cooperation and prevent abuse. Unfortunately, resourceful actors try to bypass automated filters in a bid to post content that violate platform policies and codes of conduct. To reach this goal, these malicious actors may obfuscate policy violating images (e.g. overlay harmful images by carefully selected benign images or visual patterns) to prevent machine learning models from reaching the correct decision. In this paper, we invite researchers to tackle this specific issue and present a new image benchmark. This benchmark, based on IMAGENET, simulates the type of obfuscations created by malicious actors. It goes beyond IMAGENET-C and IMAGENET-C̄ by proposing general, drastic, adversarial modifications that preserve the original content intent. It aims to tackle a more common adversarial threat than the one considered by $\ell_p$-norm bounded adversaries. We evaluate 33 pretrained models on the benchmark and train models with different augmentations, architectures and training methods on subsets of the obfuscations to measure generalization. Our hope is that this benchmark will encourage researchers to test their models and methods and try to find new approaches that are more robust to these obfuscations.

## 1 Introduction

Advances in in computer vision have lead to classifiers that nearly match human performance in many applications. However, while the human visual system is remarkably versatile in extracting semantic meaning out of even degraded and heavily obfuscated images, today's visual classifiers significantly lag behind in emulating the same robustness, and often yield incorrect outputs in the presence of natural and adversarial degradations. This is why evaluating robustness [1, 2] and improving the robustness of visual classifiers has been the subject of considerable research [3, 4], with multiple benchmarks looking at out of distribution examples [5, 6], distribution shift [7–9] or focusing on robustness to "natural degradations", like blur and noise, which are inherent in the imaging process [10, 11]. However, with more and more visual classifiers being deployed in real systems, robustness to adversarial perturbations—deliberate changes to an image introduced by an adversary to fool classifiers—has emerged as an important research direction [12–14].

Such work on adversarial robustness has largely focused on perturbations that are imperceptible to human observers—often cast as an explicit $\ell_p$-norm constraint between the original and perturbed

image [12, 15]. However, for many visual classifiers that are focused on keeping offensive, dangerous, pirated, or other policy-violating content off of online platforms, this limited definition does not suffice. In the real world, attackers who want to hide malicious content in images are not prevented by the fact that an image appears obviously perturbed to human observers, as long as the observer is also able to glean the underlying violating content [16]. This scenario is not theoretical—such malicious image manipulations are being used today by bad actors daily and at scale on modern online platforms [17].

Our work focuses on enabling research into making visual classifiers robust to such adversarial obfuscations—image transformations that can fool classifiers while leaving the underlying semantic content intelligible to human observers, but differing from prior adversarial attacks [12, 15] in allowing it to be obvious that the image is transformed. To this end, we introduce a benchmark to characterize the performance of classifiers on obfuscated images.

Naturally, we do not claim this benchmark to be exhaustive, since the space of adversarial obfuscations is limited only by the creativity of attackers and the resilience of the human visual system in inferring the underlying content despite significant manipulation. However, to provide a concrete starting point to make and measure progress in obfuscation robustness, we introduce a set of 22 transforms, illustrated in fig. 1, that (a) are compositions of various transformations available in image editing software—geometric and color transformations, image splicing and blending, style transfer, etc.; (b) are strong enough to fool most current classifiers while still leaving the underlying semantic content identifiable; and (c) are diverse enough to allow us to measure generalization of current and future robust training techniques, by measuring performance on obfuscations that are held out during training. Moreover, these transforms are similar to actual obfuscations that we have observed being used in the wild by attackers attempting to bypass Google's image policy classifiers.

Overall, our main contributions are as follows:

- We create, curate and tune a set of 22 strong, diverse, adversarial obfuscations. Compared to other benchmarks our obfuscations imitate methods by bad actors trying to fool content filter models and are allowed to drastically change the images. Our benchmark is set up to have training and hold-out obfuscations which allows to measure generalization to unknown obfuscations and monitor progress of leveraging known attacks.
- We evaluate 38 different pretrained models on our benchmark and train additional models to compare the effects of 7 different augmentation and 4 distribution shift algorithms. Our experiments show that scaling architectures, pretraining on larger datasets and choosing the right augmentations can make models more robust to unseen obfuscations.
- We train models on over 60 subsets of our training obfuscations to show relationships between them, which of them have the biggest effect on generalization and that we get diminishing returns when adding obfuscations to the training set.
- Finally, we show that training on our training obfuscations increases performance on all of 8 IMAGENET variants and that $\ell_p$-norm based adversarial training does not help robustness to our obfuscations.

We expect that our analysis, and particularly, this benchmark will stimulate research in the development of training techniques that make classifiers more robust to adversarial obfuscations. Such techniques promise to be of immediate practical value since they can be used by online platforms to strengthen their automated classification and detection models, thereby helping improve their users' online experience by keeping out unsafe and illegal content.

## 2 Related Work

**Datasets of natural distribution shifts.** Characterizing model failures and empirically estimating their consequences often requires collecting and annotating new datasets. Hendrycks et al. [6] collected datasets of natural adversarial examples (IMAGENET-A and IMAGENET-O) to evaluate how model performance degrades when inputs have limited spurious cues. Hendrycks et al. [9] collected real-world datasets (including IMAGENET-R and DEEPFASHION REMIXED) to understand how models behave under large distribution shifts such as artistic renditions of various objects. Particular shortcomings can only be explored using synthetic datasets [18]. Hendrycks and Dietterich [19] introduced IMAGENET-C, a synthetic set of common corruptions.

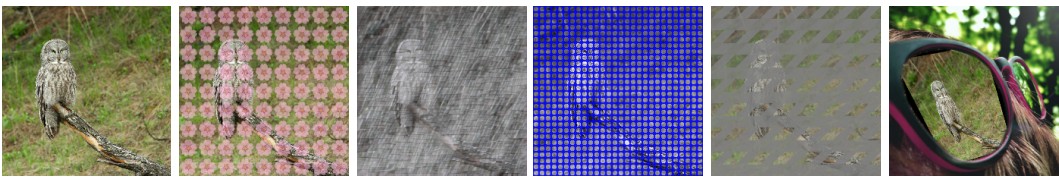

Figure 1: Example images for obfuscations. From top left to bottom right: *Clean*, *IconOverlay*, *Texturize*, *ColorPatternOverlay*, *LowContrastTriangles*, *PerspectiveComposition*. See section appendix A.1 for examples of all obfuscations.

Recently, [11] proposed IMAGENET-C̄ in a bid to understand whether progress on IMAGENET-C is truthful. They sample corruptions that are perceptually dissimilar from IMAGENET-C in feature space and observe that data augmentations may not generalize well.

Extending upon [19], Kar et al. [20] introduce corruptions that capture 3D information. They aim to guard against natural corruptions, such as camera rotation, camera focus change, motion blur.

All of the previously mentioned datasets either collect natural out-of-distribution examples or create variations that mimic natural corruptions that occur when capturing images. Our benchmark dataset goes beyond the realistic corruptions considered by above work and includes artificial corruptions that adversaries could produce using common image editing software.

Some examples of datasets that feature less natural transformations are Geirhos et al. [5], who study the propensity of Convolutional Neural Networks (CNNs) to over-emphasize texture cues, by evaluating such models on 4 types of obfuscations focused on texture and shape, as well as a dataset with texture-shape cue conflict. Xiao et al. [8], Sagawa et al. [21] investigate whether models are biased towards background cues by compositing foreground objects with various background images (IMAGENET-9, WATERBIRDS). While these not necessarily try to simulate natural processes, they are focused mostly on understanding how models deal with changes in specific aspects of images.

$\ell_p$**-norm adversarial robustness.** In some cases, it is possible to discover failures via optimization or brute-force search. $\ell_p$-norm adversarial attacks introduce small, imperceptible perturbations to input examples, with the aim of causing misclassifications [2, 12]. While the majority of the attacks in the literature assume white-box access (i.e., all model weights are available to the attacker) [12, 2, 22–28], another line of work considers the more practical black-box setting where the attacker has no or limited knowledge about the model [29–31]. While there is a lot of merit in investigating white- or black-box adversarial robustness, most real world attacks on machine learning models are not of this form [16] which is why our benchmark tries to include obfuscations that are or realistically could be used by attackers.

**Beyond $\ell_p$ robustness.** Given the practical implications of $\ell_p$ robustness, there has been growing interest in studying broader threat models that allow for large, perceptible perturbations to images. Examples include robustness to spatial transformations [32, 33] and adversarial patches [34–38]. The majority of the work in this category considers local or simple transformations that retain most of original pixel content. Our benchmark considers a wider range of transformations, including ones that cause significant visual changes but do not change the image label.

## 3   Benchmark

### 3.1   Dataset

We choose to base our benchmark on the IMAGENET Large Scale Visual Recognition Challenge 2012 (ILSVRC2012)[1] dataset [39]. It has been established in the community as the main benchmark for image classification and there already exist several variations of it that aim to measure different aspects of robustness. This allows us to easily evaluate models trained on IMAGENET and its variants

---

[1]For simplicity we will refer to it as just "IMAGENET " for the remainder of the paper.

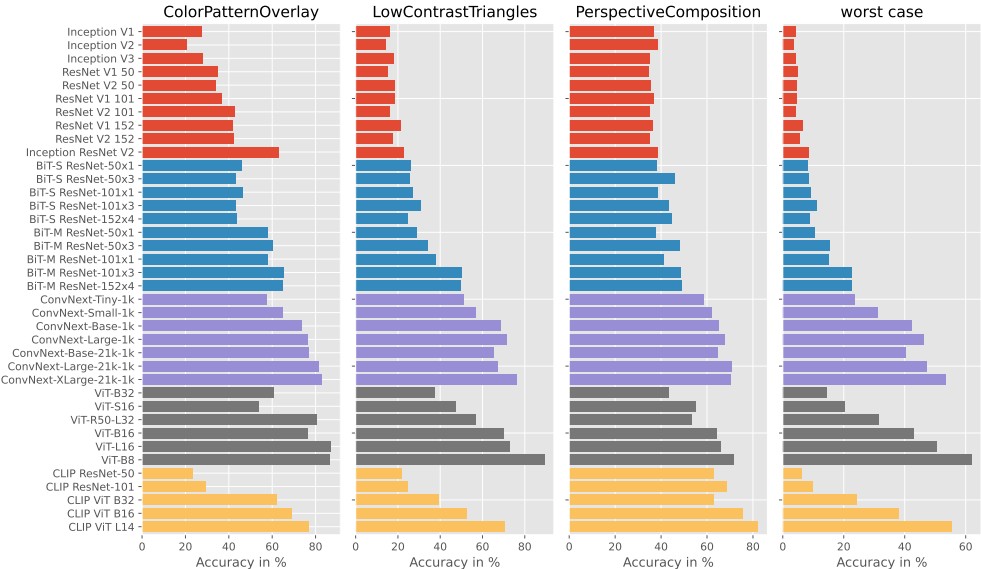

Figure 2: Accuracy on the hold-out obfuscations for multiple TensorFlow-Hub models trained on clean IMAGENET.

on our benchmark and vice versa[2]. We apply each obfuscation to each image from the IMAGENET train and validation split. As the test split includes no labels, we use the validation split as our test set, similar to IMAGENET-C [10], and use 10k images of the original training set as a validation set.

The fine grained 1000-class task of the original IMAGENET set makes it hard to apply strong obfuscations as even for humans it is hard to, e.g., distinguish over 100 different breeds of dogs. To make the classification task easier, we use the 16 super-classes introduced by [40]. They encompass 207 of the 1000 IMAGENET classes (see appendix A.7 for a detailed listing). We only do this grouping at evaluation time, which allows us to compare models trained on the standard 1000-class IMAGENET scenario. As derived in the appendix of [40] we calculate the probability of an image to be part of a super-class by using the average probability of all member classes for each super class.

### 3.2 Obfuscations

Our benchmark includes 22 obfuscations in total: 19 training obfuscations and 3 hold-out obfuscations. These represent a wide range of strong and varied manipulations covering, color changes, transformations, compositions, overlays, machine-learning based obfuscations and combinations of them. To create a static benchmark, we do not include manipulations that require access to the model prediction.

The obfuscations have a number of hyperparameters, which each has an allowed range that we randomly draw from for each image. This makes the obfuscations more diverse and avoids overfitting. We tune these hyperparameter ranges manually to get strong obfuscations, that still keep the label intact. For each obfuscation we did a grid search to find the parameters that get the worst accuracy on a Big Transfer model [41] with an underlying ResNet 152x4 pretrained on IMAGENET-21K and fine-tuned on IMAGENET-1k. We then checked visually that out of 50 example images, a maximum of 2 are not classifiable as the right super-class. If more of the labels were not recognisable we checked parameters which resulted in higher accuracy until we found a fitting set. Next, we created the range of the possible parameters around this set to get variation in the applied obfuscation but also consistent performance.

Some of the obfuscations are very scale dependent. As the original IMAGENET images are of varying sizes, we do a central crop and resizing to $224 \times 224$ before applying the obfuscations. None of

[2]The dataset and code to evaluate models can be found at `https://github.com/deepmind/image_obfuscation_benchmark`.

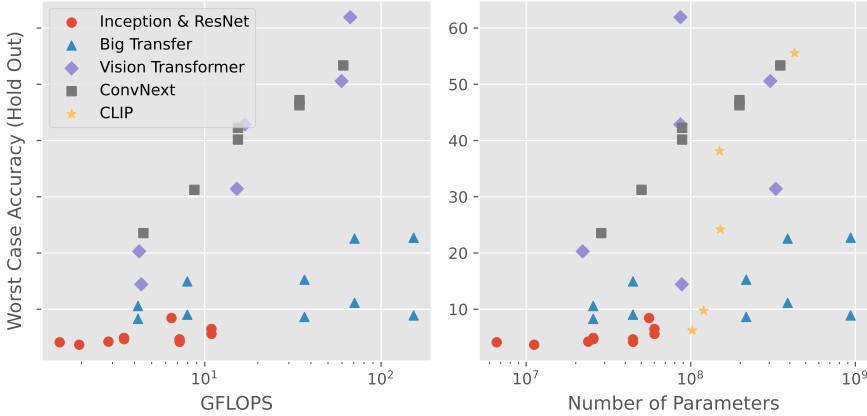

Figure 3: Worst case accuracy on hold-out obfuscations over GFLOPs (*left*, log scale) and number of parameters (*right*, log scale) for the models taken from TensorFlow-Hub. We did not include GFLOP numbers for the CLIP models as they are more than 1000x than the largest other models when doing 80 prompts for each of the 1000 IMAGENET classes.

the obfuscations change the input size, therefore all our training and evaluation data is $224 \times 224$. To stay consistent, we also do this to the clean training data, which will reduce the accuracy of the models trained for this paper as random cropping now operates on a smaller pixel space and needs to upsample.

We group the obfuscations into 5 categories: *Color Changes*, *Transformations*, *Compositions*, *Overlays*, *Machine-Learning based Obfuscations* and fig. 1 shows the hold-out obfuscations and a few examples of training obfuscations. For additional examples and a detailed description of each obfuscation, see appendix A.1 in the supplementary material.

When an obfuscation uses other images to overlay, merge or stylize the original images, we make sure that the chosen images do not introduce any objects which could be categorized into one of the 16 super-classes we evaluate on.

We chose *ColorPatternOverlay*, *LowContrastTriangles* and *PerspectiveComposition* as hold-out obfuscations because they cover multiple obfuscation categories (e.g. *ColorPatternOverlay* being both and overlay and a color change), they combine concepts from training obfuscations (*Perspective-Composition* being a combination of *PerspectiveTransform* and *PhotoComposition*) and are among the hardest obfuscations for multiple model families (*LowContrastTriangles* is the strongest for ResNet models, while *PerspectiveComposition* is the strongest for Bit, ConvNext and ViT models).

## 3.3 Metric

Merging the classes into the 16 super-classes introduces a class imbalance in the evaluation data. We counteract this by weighting the accuracy by the inverse of the number of images of that super-class, thereby giving each super-class equal contribution.

As described earlier, we have 3 hold-out obfuscations that cannot be used during training. The accuracy on these is our main metric. To simulate adversaries trying out multiple obfuscations, we combine the obfuscations on a worst-case basis, i.e., to correctly classify an image, a model has to correctly classify it under all 3 hold-out obfuscations.

Since our super-classes are very unbalanced (see appendix A.7) calculating the average accuracy across all images would be mostly determined by a model's performance on a few super-classes. As an example: Just one super-class (Dog) would account for more than half of the overall accuracy, i.e. a classifier who perfectly classifies all dogs, but no other images, would get almost 53% accuracy, a classifier that perfectly classifies dogs and birds even over 76%. Therefore when we calculate the overall accuracy, we weight each image by the inverse of the super-class occurance, i.e. we average the per-class accuracy.

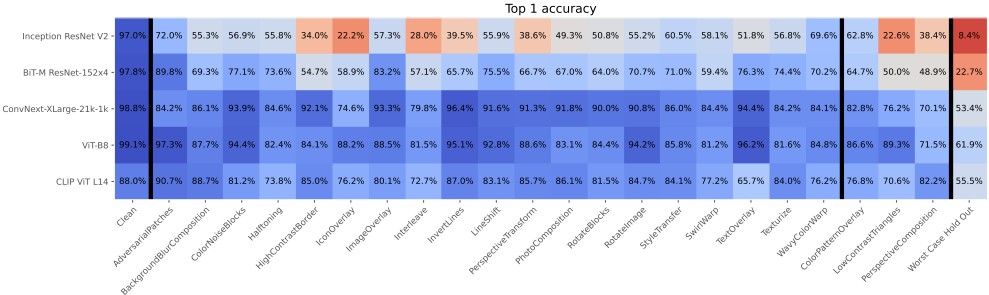

Figure 4: Top 1 accuracy for the best models from the 5 pretrained model collections.

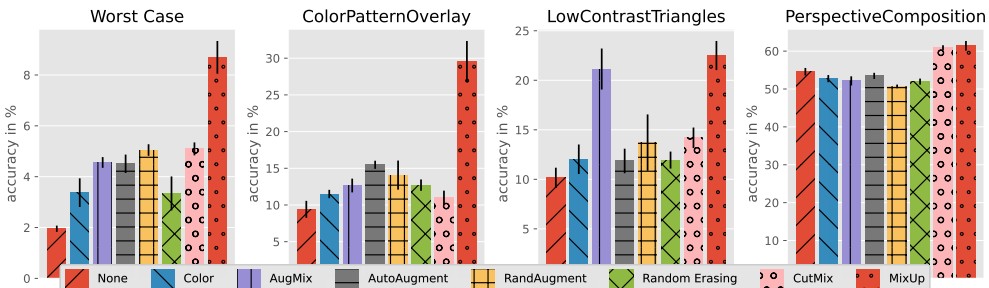

Figure 5: Accuracy on hold-out obfuscations and worst case accuracy for models trained with different augmentation schemes. All models are ResNet50 trained only on clean images. The augmentations used are color [42], AugMix [43], AutoAugment [44], RandAugment [45], random erasing [46], CutMix [47] and MixUp [48].

In summary we calculate our final metric by these steps:

1. Load the IMAGENET 2012 validation dataset
2. Filter out all images that do not share a class label with our 16 super classes
3. Obfuscate each image with the 3 hold-out obfuscations
4. For each obfuscated image evaluate the class probabilities for all 1000 IMAGENET classes
5. Calculate probabilities for each super class by averaging probabilities of all member classes
6. For each image check if the highest probability is for the correct super-class for **all** obfuscated version of that image
7. Calculate the final accuracy by averaging over all images weighted by the inverse of the super-class occurrence in the dataset

## 4 Experimental Results

We start our experiments by evaluating a range of pretrained IMAGENET classification models to analyse the robustness of different architectures to the obfuscations, and the effect of scaling models and pretraining datasets. We then compare 7 data augmentation schemes, and in section 4.3, train models on different subsets of training obfuscations to see their contribution to generalizing to the hold-out obfuscations. We then look at models trained on all training obfuscations, evaluate the effect of distribution shift algorithms, and at the end compare with results on other robustness benchmarks.

### 4.1 Evaluating Pretrained IMAGENET Models

All of the following models are evaluated on training and hold-out obfuscations as is, without any fine-tuning. We evaluate models from four different collections on TensorFlow-Hub[3]: Inception

---

[3]https://www.tensorflow.org/hub

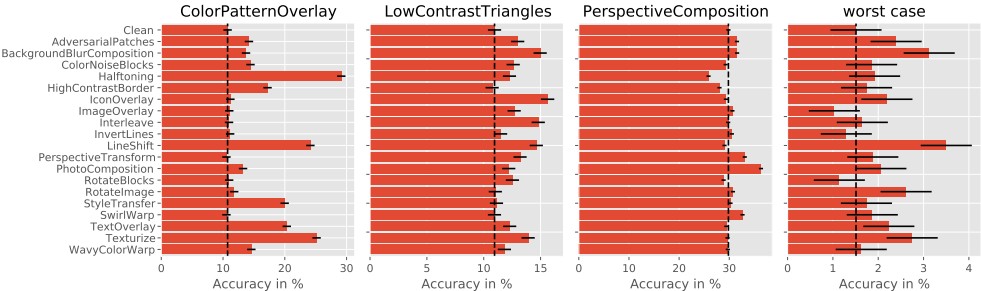

Figure 6: Accuracy on hold-out obfuscations and worst case for models trained on clean images plus a single obfuscations.

and ResNet[4], Big Transfer[5], Vision Transformer[6] and ConvNext[7]. We also include several zero-shot models based on CLIP [54], with different image encoder architectures[8]. To obtain the best performance possible for CLIP, we used the best text-prompt configuration reported by [54] for ImageNet, consisting of a combination of 80 curated prompts. The accuracy of all models on the hold-out obfuscations and the worst case accuracy for all the models is plotted in fig. 2. We see that the ViT-B8 performs the best across all models, as it does on the standard IMAGENET dataset. The results for the Big Transfer models show the gain in accuracy by pretraining on the IMAGENET-21K dataset (BiT-S models are trained on standard IMAGENET, i.e. ILSVRC2012, while BiT-M models, are pretrained on IMAGENET-21K). This can be seen in more detail in fig. C.2 in the appendix, where we see accuracy for each individual obfuscation for all BiT models.

The performance of zero-shot models exhibits an interesting behavior. On some obfuscations such as *ColorPattenOverlay* and *LowContrastTriangles* the performance of the zero-shot CLIP model is slightly worse than that of the equivalent trained model with the same vision tower. However, on *PerspectiveComposition* it actually performs better than the trained model. This behavior is probably due to the distribution of the pretraining data distribution—*PerspectiveComposition* is a more "natural" looking type of obfuscation, which might have been present in CLIP's pretraining dataset.

Figure 3 shows that when comparing models of the same type, scaling them up in terms of parameters or computation leads to increased robustness to the obfuscations. It also shows that ConvNext and ViT models outperform BiT and ResNet models. This might be due to their patchified stem that has been shown to be more robust to $\ell_p$-norm adversarial attacks [55]. Additionally, both ViT and ConvNext models are all pretrained on IMAGENET-21K. In fig. 4, we look at the results for the best model from each category over all obfuscations. While the largest vision transformer model outperforms all other models on the worst-case hold-out accuracy, it does not get the best performance across all obfuscations, e.g. the largest ConvNext model performs better on 8 of the 22 obfuscations, in some cases by a large margin (e.g. *HighContrastBorder* by 8% and *PhotoComposition* by 8.7%), indicating that different architectures are more robust to different obfuscations.

## 4.2   Comparing Augmentation Methods

One way to make models robust to out of distribution data is to use general augmentation schemes. In fig. 5, we see that all augmentations help the worst case accuracy, but only CutMix [47] and MixUp [48] improve accuracy across all 3 hold-out obfuscations, with MixUp giving by far the strongest boost overall. AugMix [43], which is one of the strongest methods on IMAGENET-C, gives a big boost on *LowContrastTriangles* but does not improve accuracy on *PerspectiveComposition*, which is very different from the natural corruptions in IMAGENET-C. This highlights that while many augmentations help with obfuscation robustness most are not universally helpful across all of them.

---

[4]https://tfhub.dev/google/collections/image based on [49] and [50].

[5]https://tfhub.dev/google/collections/bit/ based on [41].

[6]https://tfhub.dev/sayakpaul/collections/vision_transformer based on [51] and [52].

[7]https://tfhub.dev/sayakpaul/collections/convnext based on [53].

[8]https://github.com/OpenAI/CLIP

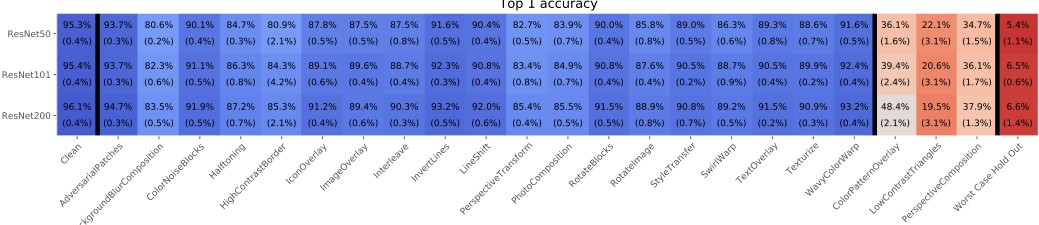

Figure 8: Top 1 accuracy for 3 different ResNet sizes when training on all training obfuscations.

## 4.3 Training on Obfuscations

In this section we train models on all or subsets of the 19 training obfuscations to analyse the effects and interactions. Unless specified otherwise, the models use a ResNet50 architecture and if error bars are plotted they represent the standard deviation from training 5 identical models with different random seeds. When we train on obfuscated images we always sample clean images with a weight of 0.5 and each obfuscation with equal weights summing up to 0.5.

### 4.3.1 Training on Subsets of Training Obfuscations

To see if there are interactions between different training obfuscations and how each of them helps with generalizing to the hold-out obfuscations, we train models only on one obfuscation (and clean data). The results can be seen in fig. 6. There are some clear connections between training and hold out obfuscations, e.g. *Halftoning*, *LineShift*, *StyleTransfer*, *TextOverlay* and *Texturize* all increasing the accuracy on *ColorPatternOverlay* significantly, while *PerspectiveTransform* and *PhotoComposition* lead to small improvements of *PerspectiveComposition*.

In appendix C.4 we also investigate the opposite, where we train models on all but one of the training obfuscations and observe similar behaviour for some obfuscations, e.g. that omitting *LineShift* significantly reduces the performance on *ColorPatternOverlay* but in other cases, like excluding *StyleTransfer*, this is compensated by the other training obfuscations.

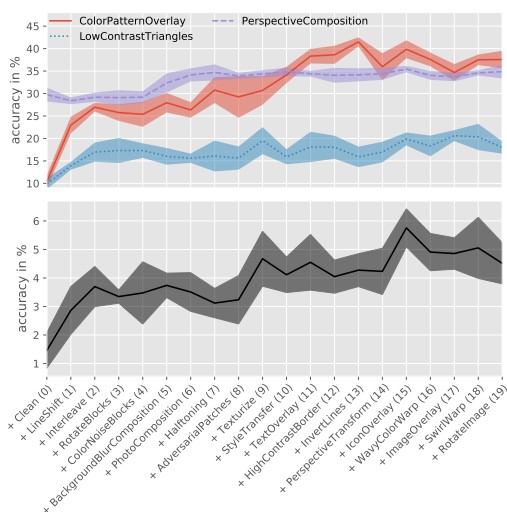

Figure 7: Accuracy on hold-out obfuscations (*top*) and worst case accuracy (*bottom*) when training on an increasing number of obfuscations.

To evaluate the effect of compounding training obfuscations, we proceed by sorting the training obfuscations by the mean accuracy on the training obfuscations when training only on images from that obfuscation. We then train models by an increasing number of obfuscations starting with the obfuscation that increases the mean accuracy the most. From the results in fig. 7 we can observe that the accuracy makes jumps when adding specific obfuscations that help with one of the hold-out obfuscations but there does not seem to be constant improvement from adding more and more obfuscations to the training data. We also observe that adding the obfuscated images to the training data does not reduce the clean accuracy of the models. We rather see a mild increase in clean accuracy when looking at fig. C.14.

### 4.3.2 Using All Training Obfuscations

In fig. 8 we see all models, even small ones, can achieve high accuracy on the training obfuscations when they see them during training. However, there is only limited generalization to the hold-out obfuscations even for the bigger models. When comparing results from fig. 2, we see that training a ResNet200 on all 19 training obfuscations is clearly outperformed by the larger BiT, ConvNext and

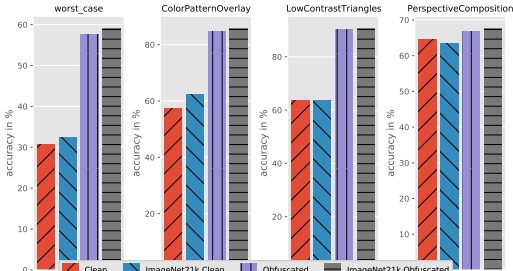

Figure 9: Comparison of a vision transformer model trained only on clean or clean and obfuscated images both with and without pretraining on the IMAGENET-21K dataset.

Figure 10: Comparing performance of different domain shift algorithms on the hold-out obfuscations, when training on all training obfuscations.

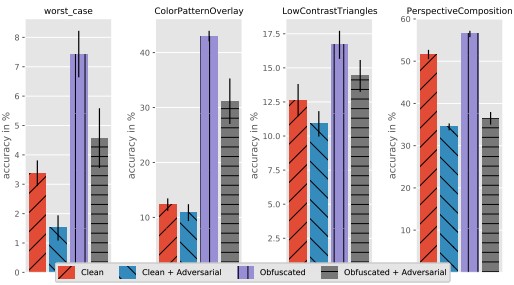

Figure 11: Hold-out accuracy of models with and without adversarial training both for training only on clean data and training on the training obfuscations. For adversarial training we used an $L_\infty$ attack with $\epsilon = 4/255$.

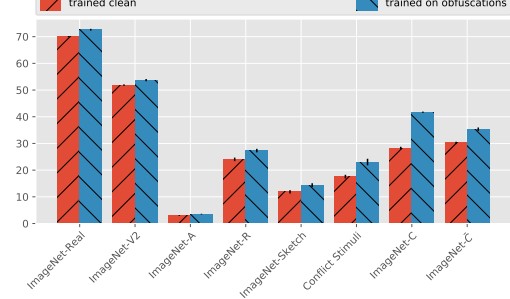

Figure 12: Comparison of a ResNet50 trained with and without the training obfuscations on multiple IMAGENET variants.

ViT models despite them never encountering any of the obfuscations during training. This shows that there is still a lot of potential to better leverage the training obfuscations for generalization.

We also train a vision transformer model both only on clean, and obfuscated images. Similar to results in section 4.1, fig. 9 shows that this achieves high accuracy on the hold-out obfuscations even without seeing obfuscations during training. Interestingly, only *ColorPatternOverlay* and *LowContrastTriangles* receive a significant boost from the addition of obfuscations to training. Pretraining on IMAGENET-21K seems to not provide significant benefits in contrast to the effect we see for the BiT models in fig. 2.

**Evaluating Algorithms Specialized for Distribution Shift** To see if we can improve generalization, we employ several approaches that were proposed to help with distribution shifts. The algorithms we evaluated are standard training using cross-entropy loss, Invariant Risk Minimization (IRM) [4], DeepCORAL [56], Domain-Adversarial Neural Network (DANN) [57], and Just Train Twice (JTT) [58]. Additional to the image and the label, these algorithms (besides JTT) are given side information about the obfuscation that has been applied to each training image. Similar to the observations in [59], fig. 10 indicates that none of the algorithms give significant improvements over the baseline.

## 4.4 Comparing to other benchmarks

In section 2, we gave an overview over existing image robustness benchmarks, many also based on IMAGENET. In this section we investigate how performance on these relates to our image obfuscation benchmark. Not surprisingly, training on our obfuscations does not give any robustness to $L_p$-norm attacks but figure fig. 11 shows that adversarial training reduces the obfuscation robustness. This is in contrast to prior observations on the effect of adversarial training for robustness to common corruptions [60] but understandable because our obfuscations often change the images drastically.

We further investigate how models trained on our training observations do on other IMAGENET variants. fig. 12 shows results for IMAGENET-Real [61], IMAGENET-V2 [62], IMAGENET-A [6], IMAGENET-R [9], IMAGENET-Sketch [63], Conflict Stimuli [64], IMAGENET-C [10] and IMAGENET-Ĉ [11]. Training on the obfuscations improves accuracy across all of these variants, however, the improvements are relatively small, indicating that our dataset represents a significantly different distribution shift than existing variants. This can also be seen in the fact that the largest improvement is seen on IMAGENET-C, as its corruptions are more similar to our obfuscations compared to the other variants.

## 5 Conclusion

In this paper, we presented a new benchmark that evaluates the robustness of image classifiers to obfuscations. To our knowledge, this is the first benchmark that curates obfuscations similar to what bad actors use to circumvent content filter models. We show that when training on obfuscations, even smaller models can achieve high robustness to them, but this does not necessarily lead to strong generalization on similar but unseen obfuscations. In our experiments, we see that newer architectures, larger models, augmentation schemes and pretraining on bigger datasets all can make models more robust to obfuscations, even if they did not have access to any during training. But there is still a gap to fill to make models robust to unseen attacks and approach human perception. We have shown that models trained on our training obfuscations also achieve better performance across multiple other robustness benchmarks. On the other hand, adversarial training does not improve obfuscation robustness. We hope this benchmark gives practitioners guidance on robustifying their models, and can drive research towards finding ways to leverage known attacks into better generalization.

### 5.1 Limitations

Investigating adversarial obfuscations leads to a large scope of design decisions. In terms of transformations, we focus on obfuscations that are independent of the model and semantic image category. This allows precomputation of obfuscated images, but also limits the space of attacks. We could further develop obfuscation methods that adapt to the image content (e.g., human-centric images may be treated differently than object-centric). In terms of data, IMAGENET is far from perfect (see e.g. [65] for an evaluation of errors made by state-of-the-art models) and has the limitation of having a single label for each image. One could generate obfuscated versions of other datasets to check the robustness of models trained in a multi-class setting or on other vision tasks like segmentation or image retrieval. The main consideration for other datasets should be to ensure that the main target label (or other output) should be very unlikely to be altered by any of the introduced obfuscations.

### 5.2 Ethical Considerations

The specific obfuscations (as in fig. 1) that we use in our benchmark may have the potential to fool automatic filters and therefore increase the amount of harmful content on digital platforms. To reduce this risk, we decided against releasing the code to create the obfuscations systematically and instead only releasing the precomputed dataset. Furthermore, our obfuscations have been tuned specifically to IMAGENET images of a fixed size. Using the same obfuscations on other images would require a reimplementation and additional tuning. It is already known that cloud based image classifiers can be bypassed with widely available transformations [66, 67]. Additionally, the type of obfuscations that we cover in our benchmark are already available in standard image editing software and it is not hard to imagine that adversaries can already think of much more elaborate approaches than the ones we have presented here. Therefore, the benefits of creating a publicly available benchmark that can help discover new methods for training robust models and set an objective baseline for the evaluation of safety far outweigh the risks of presenting examples of obfuscations on IMAGENET.

## Acknowledgments and Disclosure of Funding

We want to thank Chun-Sung Ferng, Dongjin Kwon, Sylvestre-Alvise Rebuffi and Olivia Wiles for their help with this work.

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

# A Benchmark Details

## A.1 Obfuscations

In the following section we give a description of all 18 training and 3 hold-out obfuscations in the benchmark. The hyperparameters that are drawn randomly drawn from predefined ranges for each image are *emphasized*. The obfuscations are grouped into 5 categories but these categorizations are very loose as some obfuscations, especially the hold-out obfuscations, can cover multiple concepts. Examples of the training and hold-out obfuscations can be seen in fig. A.1 and fig. A.2, respectively.

## A.2 Color Changes

**ColorNoiseBlocks**    The image is separated into blocks of a specific *block size*. Independently for each block, one of the 3 color changes is randomly selected and a uniform random value is added to that color channel. We ensure the value is between 0 and 1 by subtracting 1 from it if it goes above.

**Halftoning (technique)**    We randomly apply one of 4 different halftoning *techniques* to an image. Each *technique* reproduces the image by separating the image into blocks of equal size and replacing them by some geometric object that has a property that is proportional to the average intensity in the block. This is done for each color channel independently. The 4 *techniques* are using circles or squares with their size being proportional to the intensity, using zigzag lines whose frequency is proportional to the intensity or using random pixels whose number is proprtional to the intensity.

**InvertLines**    We go along the image in either *horizontal or vertical* lines of a specific *width*. The lines are alternatively inverted in color or left unchanged.

**LowContrastTriangles (hold-out)**    We divide the image into 3 different areas through triangles of size *scale*. In each of the 3 areas areas the *contrast is reduced* by a different factor.

## A.3 Transformation

**LineShift (horizontal, shift length, line thickness)**    Across either *horizontal or vertical* lines of a specific *width* we shift the pixels alternatively to the left and right (or top and bottom for vertical lines) by a specific *length*. We wrap around when the shift goes beyond the image borders.

**PerspectiveTransform**    We choose a set of *coordinates* around the center of each of the quadrants. Then we apply a perspective transformation moving the corners of the image to their respective *coordinate*. The rest of the image is filled with black.

**RotateBlocks**    We separate the image into blocks of a chosen *size*. Each of them is separated into 4 blocks of equal size again and their position inside the larger block is permuted by a specified *number of rotations*.

**RotateImage**    We rotate the image by a specified *angle*. Corners will be cropped while empty space is filled with black.

**SwirlWarp**    The image is transformed by a swirl warp with specified *strength*, *radius* and *center coordinates*.

**WavyColorWarp**    We transform the image with a sine wave transformation of a certain *wave length* and *amplitude* and modify the color of the image through a *change in hue*..

## A.4 Composition

**BackgroundBlurComposition**    We shrink the image by an independent *width* and *height factor*, not necessarily keeping the aspect ratio, and compose this image on top of the original image after applying a Gaussian blur of a certain *strength* on the latter.

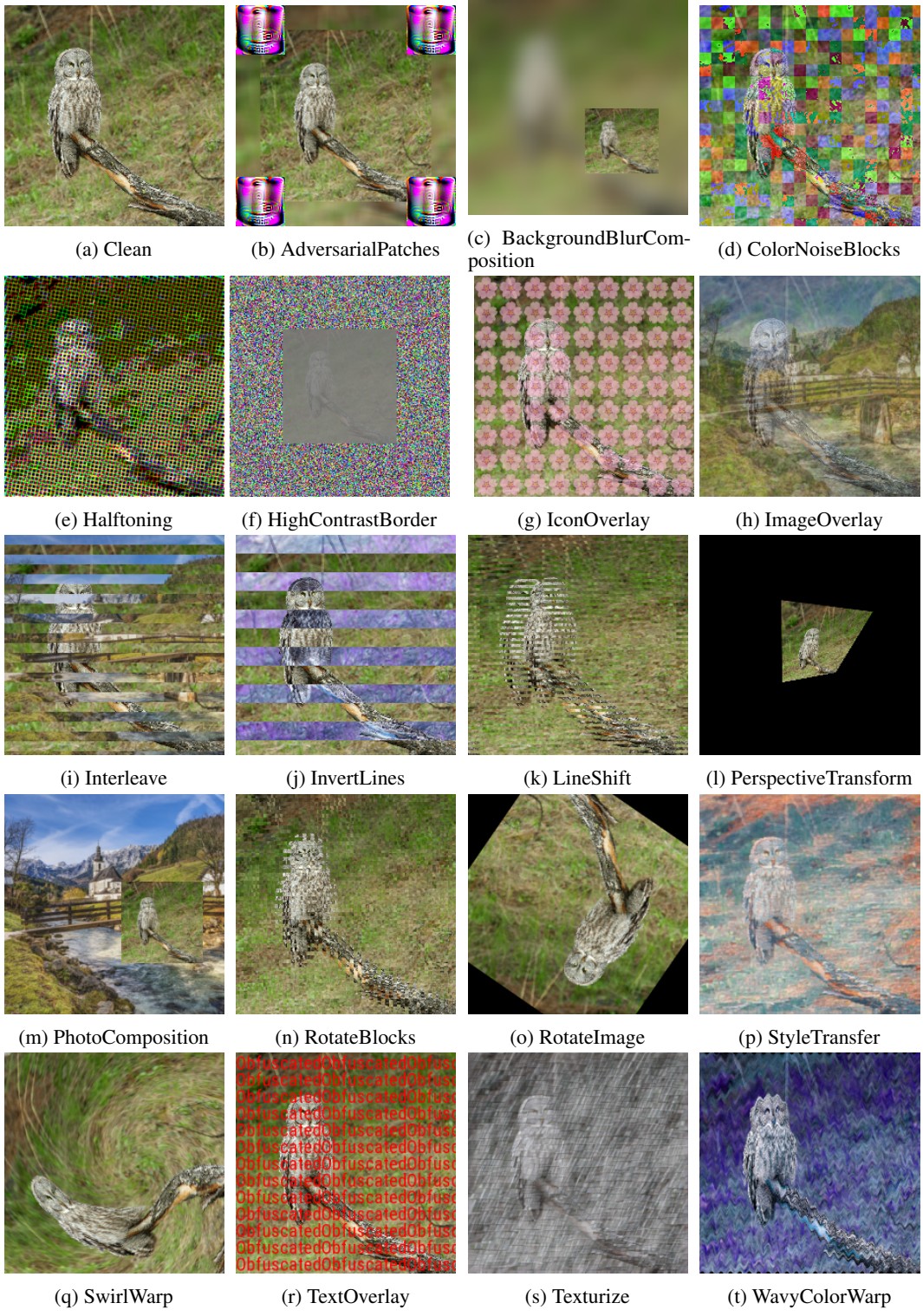

(a) Clean

(b) AdversarialPatches

(c) BackgroundBlurComposition

(d) ColorNoiseBlocks

(e) Halftoning

(f) HighContrastBorder

(g) IconOverlay

(h) ImageOverlay

(i) Interleave

(j) InvertLines

(k) LineShift

(l) PerspectiveTransform

(m) PhotoComposition

(n) RotateBlocks

(o) RotateImage

(p) StyleTransfer

(q) SwirlWarp

(r) TextOverlay

(s) Texturize

(t) WavyColorWarp

Figure A.1: Examples for the training obfuscations.

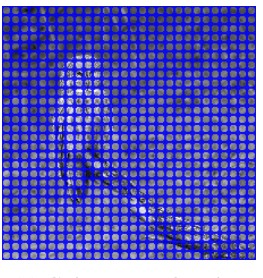

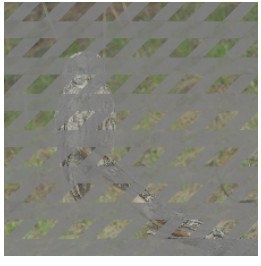

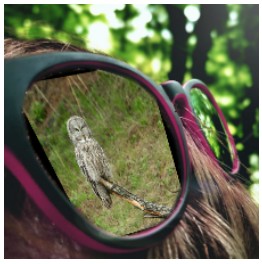

(a) ColorPatternOverlay (b) LowContrastTriangles (c) PerspectiveComposition

Figure A.2: Examples for the hold-out obfuscations.

**HighContrastBorder** The image has its *contrast reduced by a factor* then it is resized to have a border of a certain *size* around it. The border is made by sampling each pixel uniform randomly.

**PerspectiveComposition (hold-out)** We compose the image into a specific location (fixed for each *photo*). The position also defines a perspective transform that is applied to the image. Small parts of the image can also be overlayed by the *photo*. We use 14 different photos and the position that the original image is composed into has semantic meaning, e.g. it is the screen of a phone or the glass of a window.

**PhotoComposition** The image is shrunk by a *factor* and composed on top of a *photo* at a random position. There are 10 different *photos*. They are the same photos that are used for ImageOverlay and Interleave. There is no overlap with the photos that are used for PerspectiveComposition.

## A.5 Overlay

**ColorPatternOverlay (hold-out)** The image is turned into grey scale then overlayed by one of 9 *patterns* in one of 9 *colors* with a low grade of *transparency*.

**IconOverlay** A grid of a certain *number* of one of 10 *icons* is overlayed on the image with a *grade of transparency*.

**ImageOverlay** One of 10 different *photos* is overlayed over the image with a *grade of transparency*. The photos are the same that are used for ImageOverlay and Interleave. There is no overlap with the photos that are used for PerspectiveComposition.

**Interleave** The image is interleaved with one of 10 *photos*, either *horizontally or vertically*, in lines of a specific *width*. There is a low grade of *transparency*. The photos that are used are are the same as for ImageOverlay and Interleave. There is no overlap with the photos that are used for PerspectiveComposition.

**TextOverlay** One of 13 *text strings* is overlayed onto the image repeatedly in one of 9 *colors*. The *text size* is varied.

## A.6 Machine-Learning-Based Obfuscations

**Adversarial Patches** The image is shrunk and one of 3 different adversarial patches are overlayed onto the corners of the image. The patches are taken from [38].

**Stylize** We resize the image by a *factor* ($> 1$) and then choose one of 7 *classical paintings* to stylize the image. We use a model based on [68][9] for the style transfer.

---

[9]Taken from `https://tfhub.dev/google/magenta/arbitrary-image-stylization-v1-256/2`

**Texturize**  We resize the image by a *factor* ($> 1$) and then choose one of 10 *texture photos* to stylize the image. We use a model based on [68][footnote 9] for the style transfer.

## A.7  Class reduction

To allow strong obfuscations we reduce the number of classes that need to be distinguished. We follow [40] by using 16 super-classes which encompass 207 of the original 1000 IMAGENET classes.

The 16 super-classes, their IMAGENET label id and the corresponding WordNet terms are listed below. The names given to the super-classes were chosen by us.

- Airplane (1)
    - 404 (airliner)
- Bear (4)
    - 294 (brown bear, bruin, Ursus arctos), 295 (American black bear, black bear, Ursus americanus, Euarctos americanus), 296 (ice bear, polar bear, Ursus Maritimus, Thalarctos maritimus), 297 (sloth bear, Melursus ursinus, Ursus ursinus)
- Bicycle (2)
    - 444 (bicycle-built-for-two, tandem bicycle, tandem), 671 (mountain bike, all-terrain bike, off-roader)
- Bird (49)
    - 8 (hen), 10 (brambling, Fringilla montifringilla), 11 (goldfinch, Carduelis carduelis), 12 (house finch, linnet, Carpodacus mexicanus), 13 (junco, snowbird), 14 (indigo bunting, indigo finch, indigo bird, Passerina cyanea), 15 (robin, American robin, Turdus migratorius), 16 (bulbul), 18 (magpie), 19 (chickadee), 20 (water ouzel, dipper), 22 (bald eagle, American eagle, Haliaeetus leucocephalus), 23 (vulture), 24 (great grey owl, great gray owl, Strix nebulosa), 80 (black grouse), 81 (ptarmigan), 82 (ruffed grouse, partridge, Bonasa umbellus), 83 (prairie chicken, prairie grouse, prairie fowl), 87 (African grey, African gray, Psittacus erithacus), 88 (macaw), 89 (sulphur-crested cockatoo, Kakatoe galerita, Cacatua galerita), 90 (lorikeet), 91 (coucal), 92 (bee eater), 93 (hornbill), 94 (hummingbird), 95 (jacamar), 96 (toucan), 98 (red-breasted merganser, Mergus serrator), 99 (goose), 100 (black swan, Cygnus atratus), 127 (white stork, Ciconia ciconia), 128 (black stork, Ciconia nigra), 129 (spoonbill), 130 (flamingo), 131 (little blue heron, Egretta caerulea), 132 (American egret, great white heron, Egretta albus), 133 (bittern), 135 (limpkin, Aramus pictus), 136 (European gallinule, Porphyrio porphyrio), 137 (American coot, marsh hen, mud hen, water hen, Fulica americana), 138 (bustard), 139 (ruddy turnstone, Arenaria interpres), 140 (red-backed sandpiper, dunlin, Erolia alpina), 141 (redshank, Tringa totanus), 142 (dowitcher), 143 (oystercatcher, oyster catcher), 144 (pelican), 145 (king penguin, Aptenodytes patagonica)
- Boat (5)
    - 472 (canoe), 554 (fireboat), 625 (lifeboat), 814 (speedboat), 914 (yawl)
- Bottle / Jug (7)
    - 440 (beer bottle), 720 (pill bottle), 737 (pop bottle, soda bottle), 898 (water bottle), 899 (water jug), 901 (whiskey jug), 907 (wine bottle)
- Car (3)
    - 436 (beach wagon, station wagon, wagon, estate car, beach waggon, station waggon, waggon), 511 (convertible), 817 (sports car, sport car)
- Cat / Cougar (6)
    - 281 (tabby, tabby cat), 282 (tiger cat), 283 (Persian cat), 284 (Siamese cat, Siamese), 285 (Egyptian cat), 286 (cougar, puma, catamount, mountain lion, painter, panther, Felis concolor)
- Chair / Throne (4)
    - 423 (barber chair), 559 (folding chair), 765 (rocking chair, rocker), 857 (throne)
- Clock (3)
    - 409 (analog clock), 530 (digital clock), 892 (wall clock)

- Dog (109)
  - 152 (Japanese spaniel), 153 (Maltese dog, Maltese terrier, Maltese), 154 (Pekinese, Pekingese, Peke), 155 (Shih-Tzu), 156 (Blenheim spaniel), 157 (papillon), 158 (toy terrier), 159 (Rhodesian ridgeback), 160 (Afghan hound, Afghan), 161 (basset, basset hound), 162 (beagle), 163 (bloodhound, sleuthhound), 164 (bluetick), 165 (black-and-tan coonhound), 166 (Walker hound, Walker foxhound), 167 (English foxhound), 168 (redbone), 169 (borzoi, Russian wolfhound), 170 (Irish wolfhound), 171 (Italian greyhound), 172 (whippet), 173 (Ibizan hound, Ibizan Podenco), 174 (Norwegian elkhound, elkhound), 175 (otterhound, otter hound), 176 (Saluki, gazelle hound), 177 (Scottish deerhound, deerhound), 178 (Weimaraner), 179 (Staffordshire bullterrier, Staffordshire bull terrier), 180 (American Staffordshire terrier, Staffordshire terrier, American pit bull terrier, pit bull terrier), 181 (Bedlington terrier), 182 (Border terrier), 183 (Kerry blue terrier), 184 (Irish terrier), 185 (Norfolk terrier), 186 (Norwich terrier), 187 (Yorkshire terrier), 188 (wire-haired fox terrier), 189 (Lakeland terrier), 190 (Sealyham terrier, Sealyham), 191 (Airedale, Airedale terrier), 193 (Australian terrier), 194 (Dandie Dinmont, Dandie Dinmont terrier), 195 (Boston bull, Boston terrier), 196 (miniature schnauzer), 197 (giant schnauzer), 198 (standard schnauzer), 199 (Scotch terrier, Scottish terrier, Scottie), 200 (Tibetan terrier, chrysanthemum dog), 201 (silky terrier, Sydney silky), 202 (soft-coated wheaten terrier), 203 (West Highland white terrier), 205 (flat-coated retriever), 206 (curly-coated retriever), 207 (golden retriever), 208 (Labrador retriever), 209 (Chesapeake Bay retriever), 210 (German short-haired pointer), 211 (vizsla, Hungarian pointer), 212 (English setter), 213 (Irish setter, red setter), 214 (Gordon setter), 215 (Brittany spaniel), 216 (clumber, clumber spaniel), 217 (English springer, English springer spaniel), 218 (Welsh springer spaniel), 219 (cocker spaniel, English cocker spaniel, cocker), 220 (Sussex spaniel), 221 (Irish water spaniel), 222 (kuvasz), 223 (schipperke), 224 (groenendael), 225 (malinois), 226 (briard), 228 (komondor), 229 (Old English sheepdog, bobtail), 230 (Shetland sheepdog, Shetland sheep dog, Shetland), 231 (collie), 232 (Border collie), 233 (Bouvier des Flandres, Bouviers des Flandres), 234 (Rottweiler), 235 (German shepherd, German shepherd dog, German police dog, alsatian), 236 (Doberman, Doberman pinscher), 237 (miniature pinscher), 238 (Greater Swiss Mountain dog), 239 (Bernese mountain dog), 240 (Appenzeller), 241 (EntleBucher), 243 (bull mastiff), 244 (Tibetan mastiff), 245 (French bulldog), 246 (Great Dane), 247 (Saint Bernard, St Bernard), 248 (Eskimo dog, husky), 249 (malamute, malemute, Alaskan malamute), 250 (Siberian husky), 252 (affenpinscher, monkey pinscher, monkey dog), 253 (basenji), 254 (pug, pug-dog), 255 (Leonberg), 256 (Newfoundland, Newfoundland dog), 257 (Great Pyrenees), 259 (Pomeranian), 261 (keeshond), 262 (Brabancon griffon), 263 (Pembroke, Pembroke Welsh corgi), 265 (toy poodle), 266 (miniature poodle), 267 (standard poodle), 268 (Mexican hairless)
- Elephant (2)
  - 385 (Indian elephant, Elephas maximus), 386 (African elephant, Loxodonta africana)
- Keyboard / Typewriter (2)
  - 508 (computer keyboard, keypad), 878 (typewriter keyboard)
- Cleaver (1)
  - 499 (cleaver, meat cleaver, chopper)
- Rotisserie (1)
  - 766 (rotisserie)
- Van / Truck (8)
  - 555 (fire engine, fire truck), 569 (garbage truck, dustcart), 656 (minivan), 675 (moving van), 717 (pickup, pickup truck), 734 (police van, police wagon, paddy wagon, patrol wagon, wagon, black Maria), 864 (tow truck, tow car, wrecker), 867 (trailer truck, tractor trailer, trucking rig, rig, articulated lorry, semi)

# B  Experimental Details

**Training Details**  Unless other specified we trained a ResNet50 model for 300 IMAGENET equivalent epochs, with 50% of the training data clean (central cropped and resized to 224 × 224) and

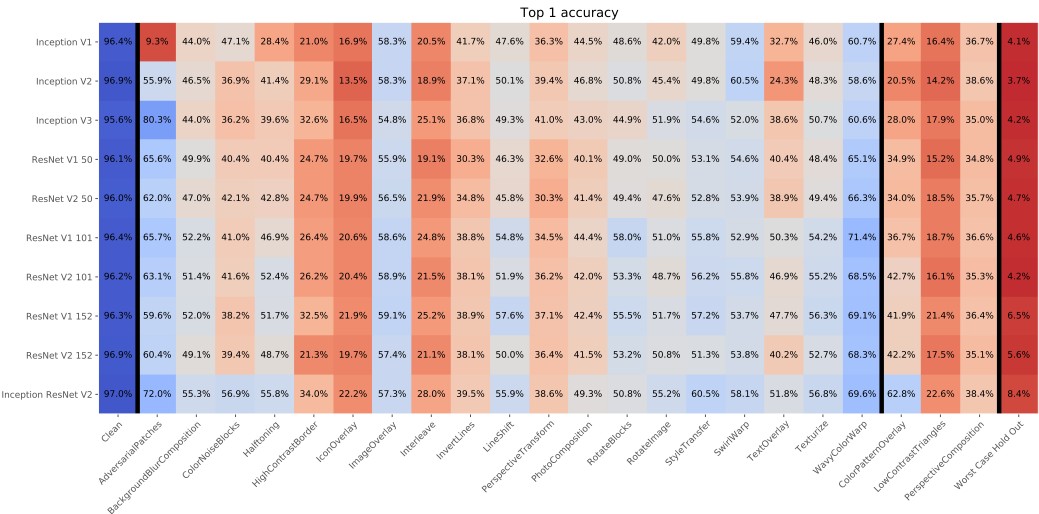

Figure C.1: Heatmap of top accuracy for all the obfuscations for multiple Inception and ResNet variants.

the rest being sampled with equal probability from the training obfuscations or a subset of them. As augmentations random crop and color augmentations as in [42] were used.

We trained models in two different frameworks with slightly different parameters. Experiments on the augmentations (fig. 5), adversarial training (fig. 11), the ImageNet variants (fig. 12) and the vision transformers (fig. 9) used framework 1. For ResNet we used SGD with momentum 0.9 and a cosine learning rate decay with base learning rate $4e^{-4}$ after linear ramp-up of 10 epochs and a batch size of 4096.

For the ViTs we used a ViT-B16 model with a weight decay of 0.3, label smoothing of 0.1, an exponential moving average with momentum 0.9999. We used the AdamW optimizer [69] with momenta $\beta_1 = 0.9$, $\beta_2 = 0.95$ and base learning rate $1e^{-4}$ with the same learning rate schedule as the ResNet50 above. For data augmentations we used random crops, as well as MixUp, CutMix and RandAugment, the latter using 2 layers, magnitude 9 and a random probability of 0.5.

Results on all the other experiments used framework 2 which used ADAM optimizer with learning rate $1e^{-3}$ which we chose after comparing experiments with learning rate $1e^{-4}$ and learning rate $1e^{-2}$ and a batch size of 512.

**Adversarial Training** For results in fig. 11 we used adversarial training that used untargeted $\ell_\infty$ attacks with $\epsilon = 4/255$. We follow [70] by creating the adversarial perturbations through 2 step projected gradient descent (PGD$^2$) with a step size of $\frac{5}{8}\epsilon \approx 0.01$. We used early stopping to avoid robust overfitting. In case obfuscated images were used the attacks were done on both clean and obfuscated images.

## C  Additional Experimental Results

To give more insights into the experiments we add more detailed figures, including both training and hold-out obfuscations, and new analysis for experiments in the main paper.

### C.1  Evaluating Baseline Models Trained on Clean IMAGENET

In section 4.1 we showed results from models taken from TensorFlow hub that were trained on the standard IMAGENET dataset (and pretrained on IMAGENET-21K in some cases). Figure 2 showed the performance of of all the models, but only on the hold-out obfuscations, while fig. 4 included the accuracy on all obfuscations, training and hold-out, but only for the best models from each type. In figs. C.1 to C.4 we show the results on all obfuscations for Inception and ResNet, BiT, ConvNext and

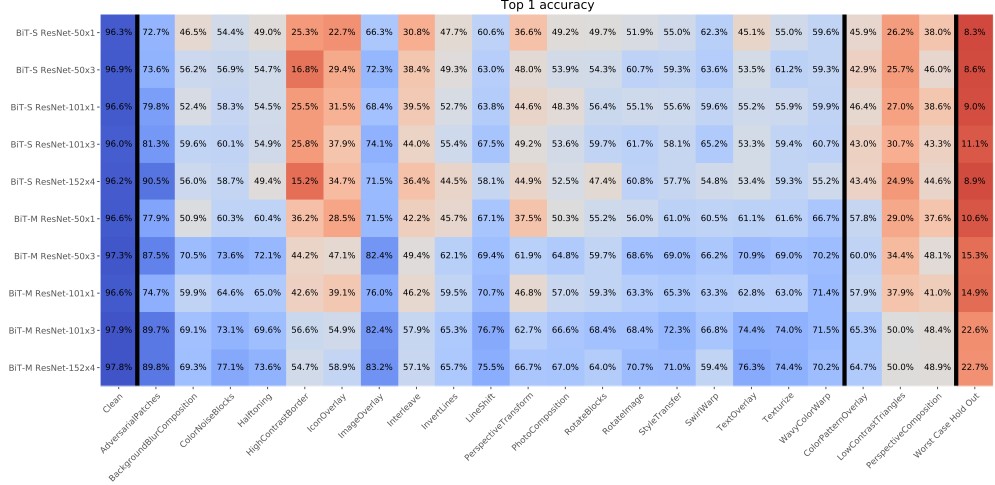

Figure C.2: Heatmap of the top 1 accuracy for all the obfuscations for multiple BiT variants.

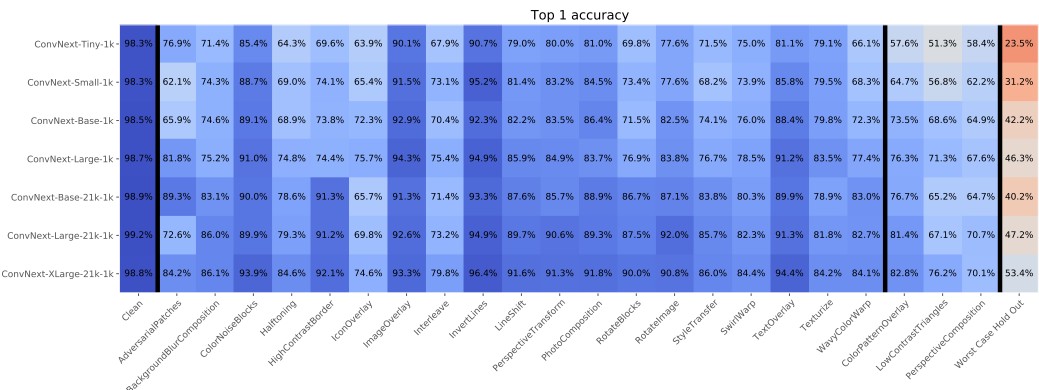

Figure C.3: Heatmap of top 1 accuracy for all the obfuscations for multiple ConvNext variants.

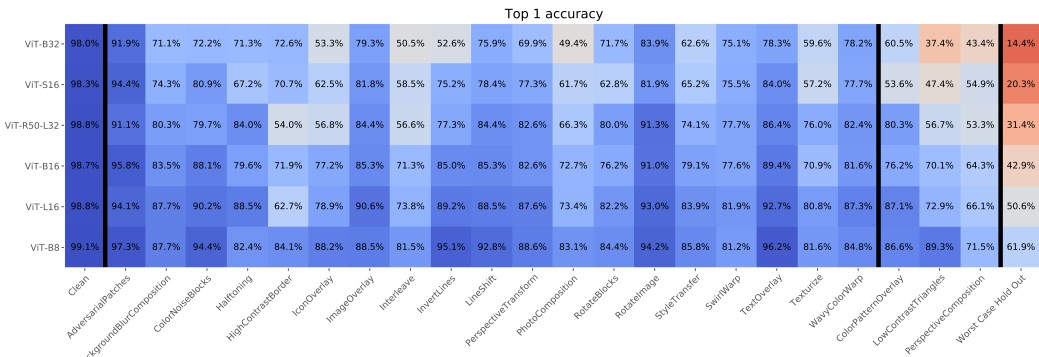

Figure C.4: Heatmap of top 1 accuracy for all the obfuscations for multiple ViT variants.

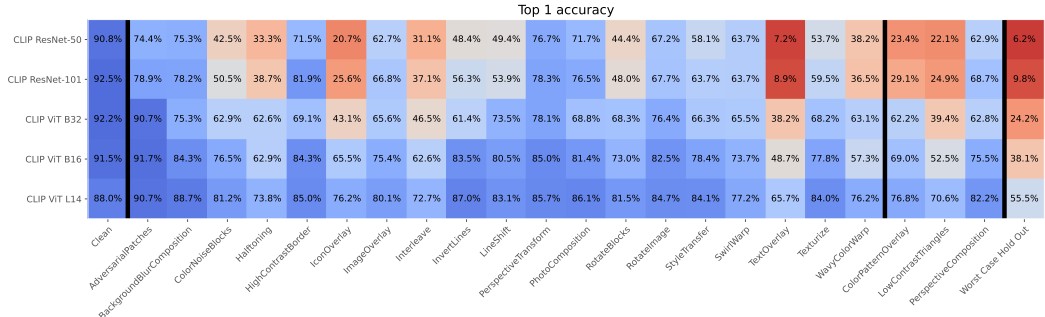

Figure C.5: Heatmap of top 1 accuracy for all the obfuscations for multiple CLIP models using 80 prompts.

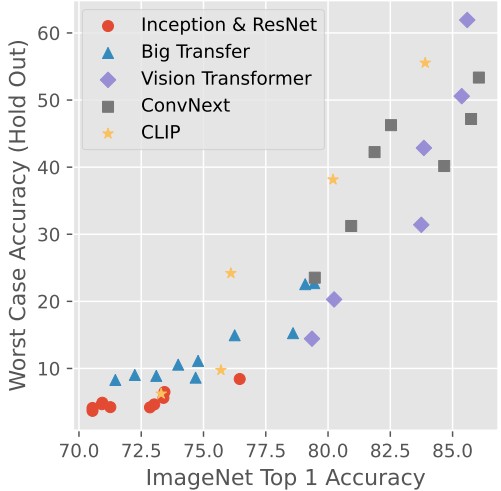

Figure C.6: Worst case accuracy on the hold-out obfuscations over accuracy on standard ImageNet (using all 1000 classes) for all the models taken from TensorFlow Hub.

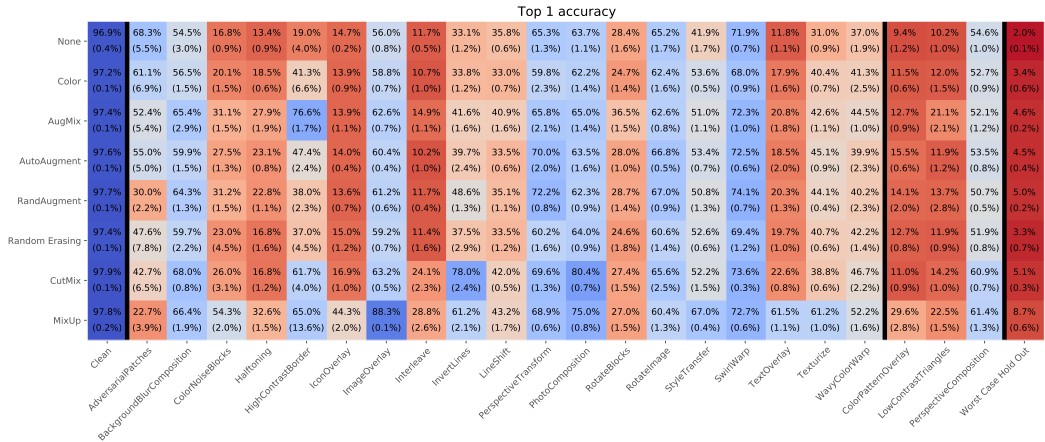

Figure C.7: Heatmap of the top 1 accuracy for all the obfuscations for cleanly trained ResNet50 models with different augmentation schemes.

ViT models, respectively. It allows a more detailed look in how robustness to different obfuscations evolves when models are scaled up or are pretrained on IMAGENET-21K.

In fig. C.1 we can see that the different Inception and ResNet models get almost the exact same accuracy on some obfuscations like *ImageOverlay* or *InvertLines*, while other obfuscations, e.g. *Halftoning* or *RotateImage*, are easier to classify for the bigger models. Interestingly, this does not hold for the other model types as we see both *ImageOverlay* and *InvertLines* get improved performance from larger models in figs. C.2 to C.4.

We can also see in figs. C.2 and C.3 how pretraining on IMAGENET-21K helps make models more robust to the obfuscations. When comparing the BiT-S models to their corresponding BiT-M models we see an improvement on all obfuscations while for ConvNext-Base and ConvNext-Base-21k models this is more mixed, e.g. *IconOverlay Interleave*, *LowContrastTriangles* and *PerspectiveComposition* show no improvement, mirroring our results in fig. 9.

We mentioned in section 4.1 that the models that perform strongest on our hold-out obfuscations are also have the highest accuracy on the standard IMAGENET benchmark. This is visualised in fig. C.6 showing a clear correlation between the accuracy on the 1000-class accuracy on standard IMAGENET and the worst-case accuracy on our hold-out obfuscations.

In fig. C.7 we can see how the augmentation schemes that we evaluated in the main paper perform on all the obfuscations. As in fig. 5 we see that MixUp performs best, increasing the accuracy significantly for 15 of the 19 training obfuscations. You can also see clear connections between the augmentation methods and the obfuscations they do particularly well on. For example, MixUp does best on *IconOverlay*, *ImageOverlay* and *TextOverlay*, while CutMix does best on *InvertLines*, *BackgroundBlurComposition* and *PhotoComposition*.

## C.2    Evaluating Upper Ceiling for Benchmark Metric

While we tried to carefully tune our obfuscations to be strong *and* keep the semantic content of the images intact, it is inevitable that there will be cases where the image is obfuscated beyond recognition. This is not unusual, even the clean IMAGENET dataset contains wrong labels and really hard images that even humans fail to label correctly [71]. To evaluate the upper ceiling of performance on our benchmark we create an oracle by combining multiple models in a best case way, i.e. we use the predictions of the ViT B8 model from fig. C.4 for *ColorPatternOverlay* and *LowContrastTriangles* and the predictions of the CLIP ViT L14 from fig. C.5 for *PerspectiveComposition* as these models get the best results on these hold-out obfuscations. Doing this we get a worst case hold out accuracy of 69.82% which is almost 8% higher than the best model (the ViT B8) gets. If we go further and combine both these models on an image-by-image basis, i.e. we give the correct prediction if at least one of the models predicts the correct super class. Doing this yields a worst case hold out accuracy of 80.36%. If we extend this to the best model from each of the 5 pretrained groups (Inception ResNet V2, BiT-M ResNet-152x4, ConvNext-XLarge-21k-1k, ViT B8 and CLIP ViT L14) we get to 86.89% and when using all 38 pretrained models we get 95.12% although the later might be discarded as too extreme as obviously we could get arbitarily close to 100% by combining more and more random models. Nontheless, we think this analysis shows that results over 80%, and therefore roughly 20% higher than what the best models we evaluated achieved, are realistic for the hold out worst case accuracy.

## C.3    Evaluating Choice of Hold Out Obfuscations

As described in section 3.2 we chose *ColorPatternOverlay*, *LowContrastTriangles* and *PerspectiveComposition* as hold-out obfuscations because of their conceptual diversity and difficulty. To investigate how our choice compares we plot the histogram of the worst case hold out accuracy over all 1540 possible combinations to chose 3 out of 22 obfuscations as hold out. As you can see from figs. C.8 to C.12, our choice is always amongst the lowest in terms of the worst case hold out accuracy, which underlines that we chose one of the hardest sets of hold out obfuscations. The only exception for this seems to be the CLIP models, which is most likely because, as seen in fig. C.5, *PerspectiveComposition* is one of the easier transformations for it.

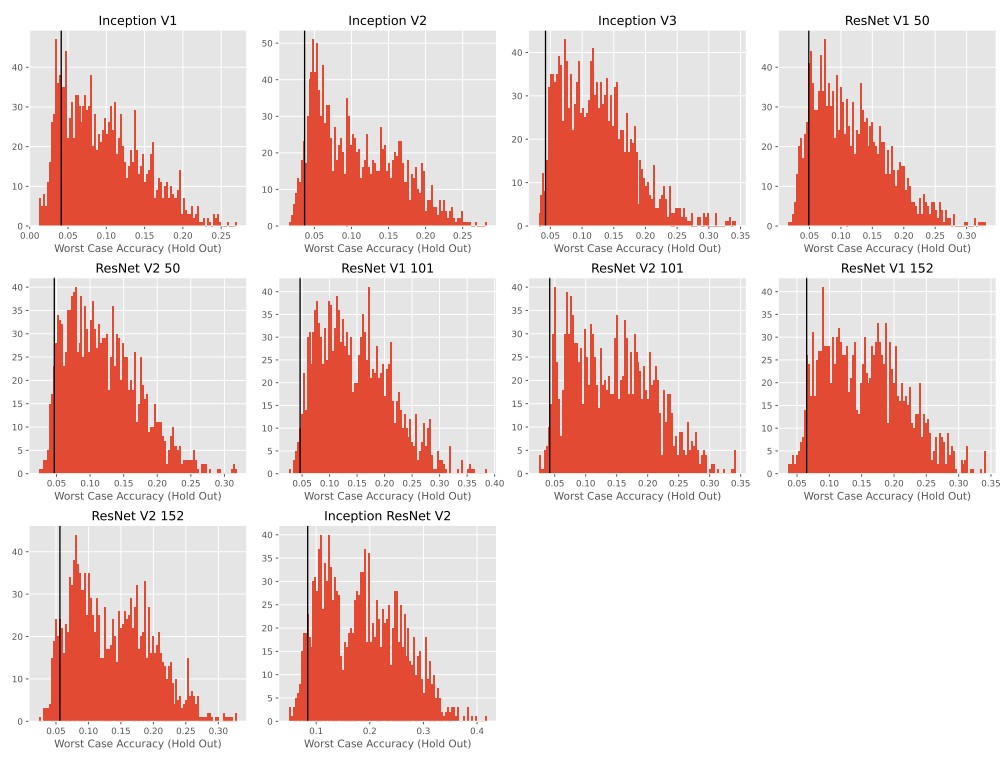

Figure C.8: Histogram of worst case hold out accuracy for all 1540 possible choices of 3 hold out obfuscations for the ResNet models. Our choice is marked with a vertical black line.

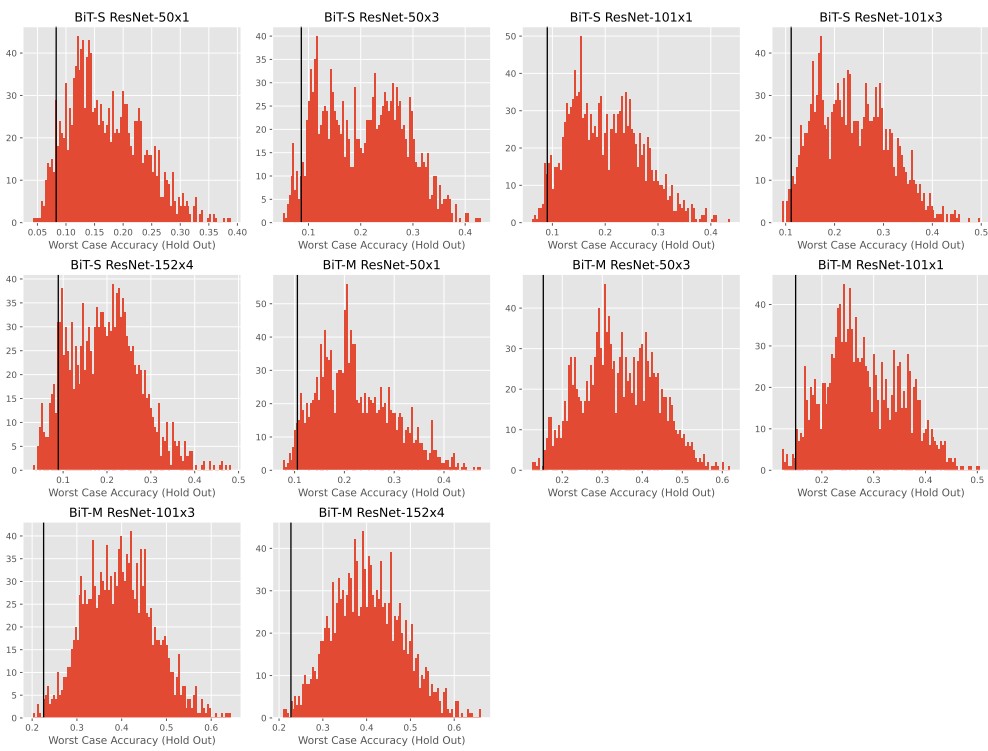

Figure C.9: Histogram of worst case hold out accuracy for all 1540 possible choices of 3 hold out obfuscations for the BiT models. Our choice is marked with a vertical black line.

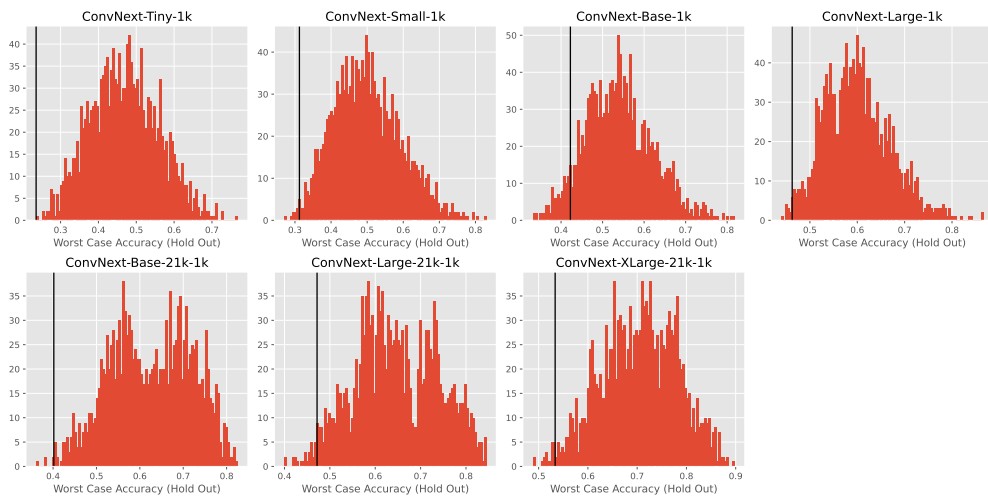

Figure C.10: Histogram of worst case hold out accuracy for all 1540 possible choices of 3 hold out obfuscations for the ConvNext models. Our choice is marked with a vertical black line.

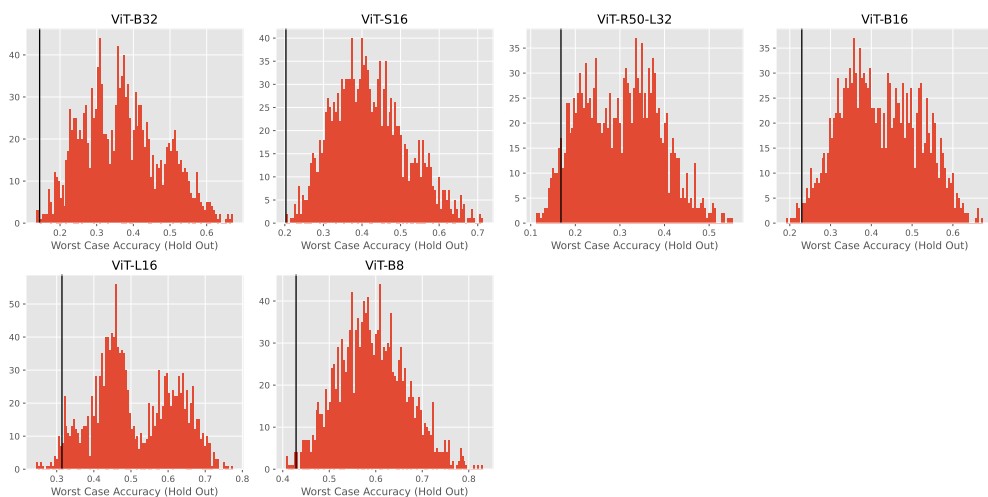

Figure C.11: Histogram of worst case hold out accuracy for all 1540 possible choices of 3 hold out obfuscations for the ViT models. Our choice is marked with a vertical black line.

## C.4  Evaluating Models Trained on Subsets of Training Obfuscations

The results for all obfuscations when training on a single obfuscation (and clean images) are show in fig. C.13, corresponding to the hold-out results shown in fig. 6. The heatmap allows us to see relationships between training obfuscations, e.g. training on *Interleave* giving strong boosts to the accuracy on *InvertLines* and *LineShift* or training on *RotateImage* improving robustness to *SwirlWarp*. Sometimes this relationships is not symmetric e.g. while training on *PhotoComposition* gives a very large boost of over 30% on *BackgroundBlurComposition* the reverse only improves performance by less than 10 %.

In fig. C.14 we see what happens to robustness to each obfuscation when adding more and more obfuscations to the training data. In fig. 7 in the main paper we saw the the hold-out performance has jumps when certain obfuscations are added but stays mostly flat otherwise. Here we can see are more nuanced picture where there is some clear generalization to other obfuscations initially but when adding obfuscations like *TextOverlay* later on there is no significant increase in the performance of other training obfuscations. This is not surprising as the obfuscations to add to the training set were ordered by their average performance across the training obfuscations in fig. C.13.

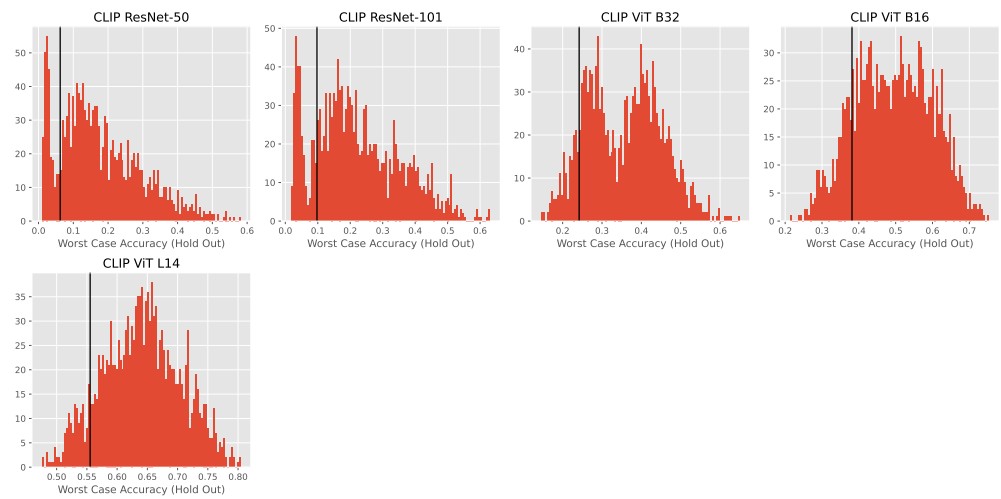

Figure C.12: Histogram of worst case hold out accuracy for all 1540 possible choices of 3 hold out obfuscations for the CLIP models. Our choice is marked with a vertical black line.

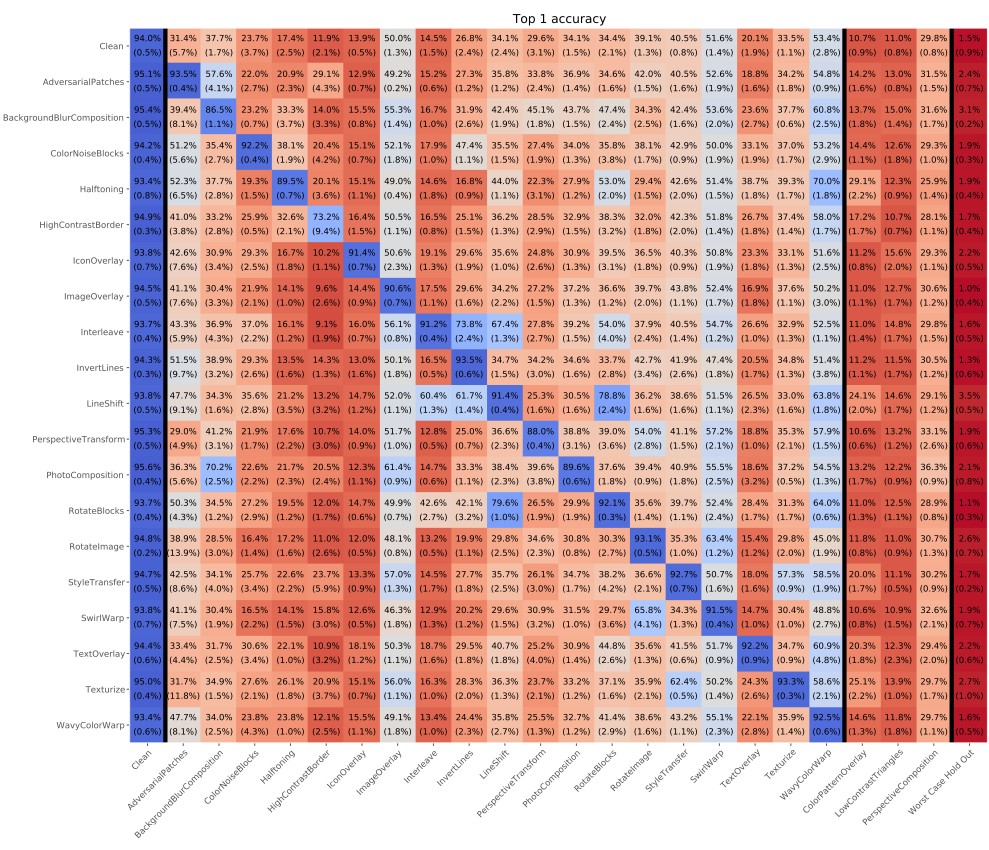

Figure C.13: Heatmap of top 1 accuracy for all the obfuscations for models trained on clean images plus a single obfuscations.

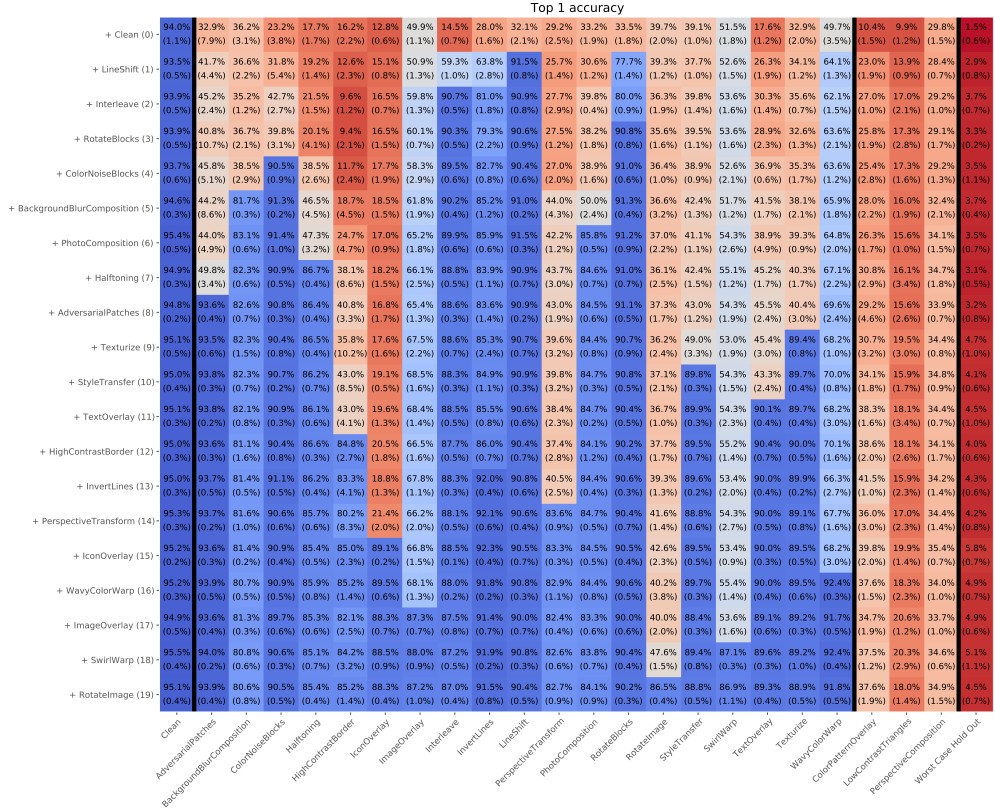

Figure C.14: Heatmap of top 1 accuracy for all the obfuscations for models trained on an increasing number of obfuscations.

fig. C.15 shows accuracy across obfuscations when training on all but one of the training obfuscations. The results on the diagonal show which of the obfuscations can be generalized on from training on the others. Similar to the results on the hold-out obfuscations in fig. 8 we see that generalization to unseen obfuscations is varied and in some cases, like *IconOverlay*, extremely limited for ResNet models.

## C.5 Comparing Other Models and Training Approaches

In fig. C.16 we see the results on all obfuscations when training with the different distribution shift algorithms from fig. 10. fig. C.17 shows the same heatmap for the ViT models we trained and whose hold-out accuracy we plotted in fig. 9. Similar to the results from the ViT models we took from Tensorflow Hub (fig. C.4) we can see that even when not training on any of the obfuscations, ViTs can achieve solid performance on most obfuscations. We also again see only small gains from pretraining on IMAGENET-21K, in contrast to the results we saw when comparing BiT and ConvNext models with and without pretraining. One possible explanation might be that pretraining only has a strong affect when performance is lower than what our ViT achieves already.

Another interesting observation is that the ViTs trained on all the training obfuscations achieve even higher performance than the ResNets we trained for fig. 8. All training obfuscations get an accuracy above 90% most even above 95%, showing that we were successful in tuning our obfuscations to keep the label intact for almost all images.

We see a more detailed view of the effects of adversarial training on the robustness to obfuscations in fig. C.18. This is based on the same experiments shown in fig. 11 but we can see that while adversarial training did not improve the accuracy on any of the hold-out obfuscations, it does offer significantly improved performance on some of the training obfuscations when training only on

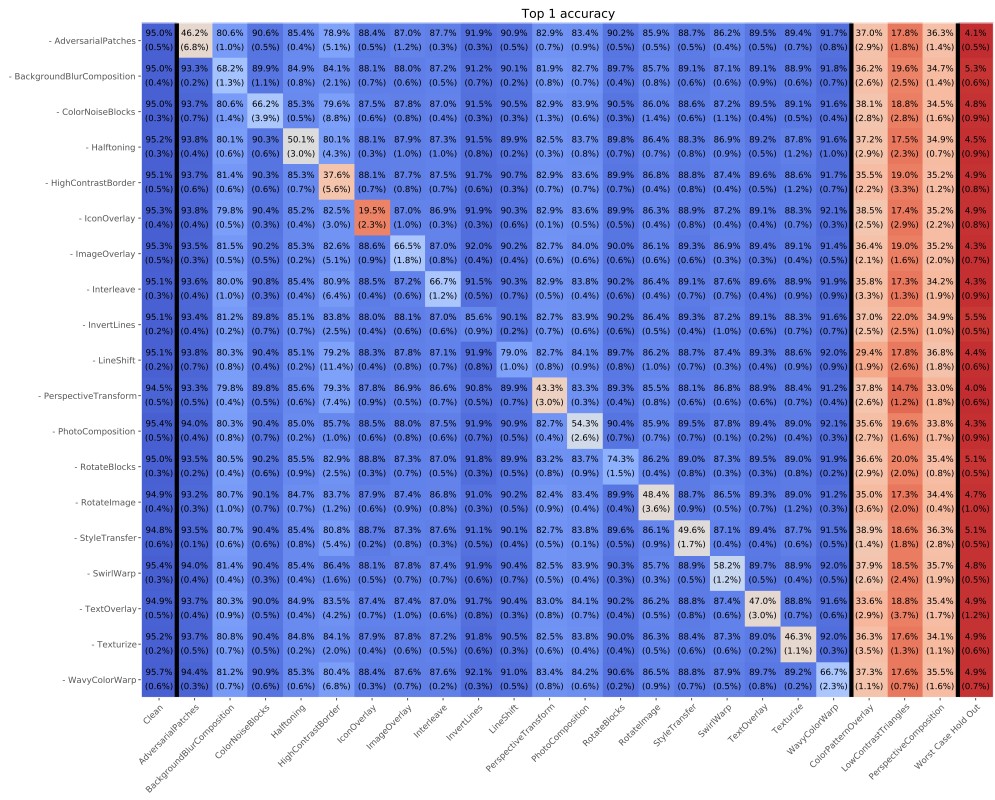

Figure C.15: Heatmap of top 1 accuracy for all the obfuscations for models trained on all but one training obfuscation.

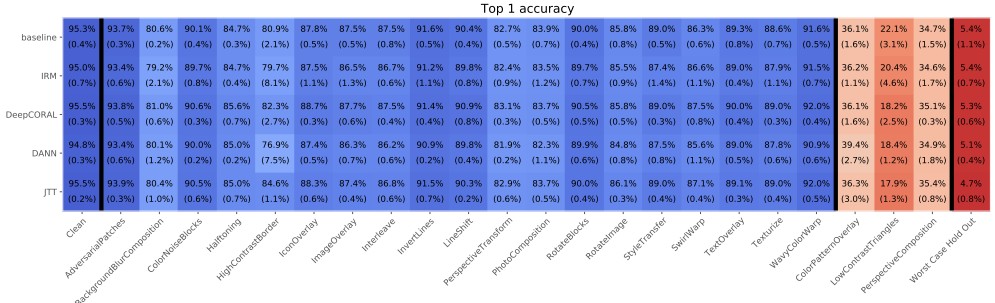

Figure C.16: Heatmap of top 1 accuracy for all the obfuscations for models trained on different distribution shift algorithms.

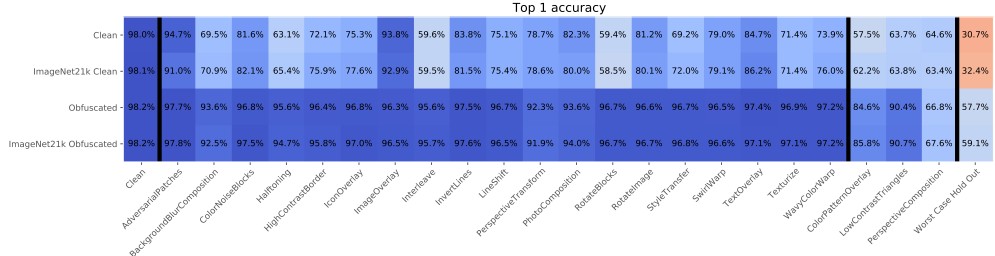

Figure C.17: Heatmap of top 1 accuracy for all the obfuscations for ViT-B16 model trained on clean images only or on the training obfuscations with and without pretraining on IMAGENET-21K

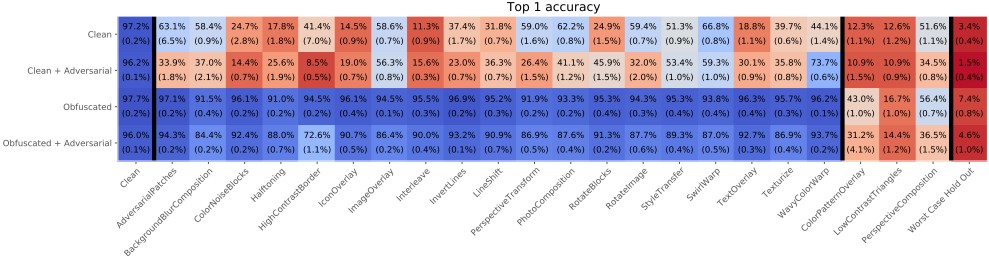

Figure C.18: Heatmap of top 1 accuracy for all the obfuscations for models trained on clean images only or on the training obfuscations with and without adversarial training.

clean images, namely *Halftoning*, *IconOverlay*, *Interleave*, *LineShift RotateBlocks*, *TextOverlay* and *WavyColorWarp*.

## C.6    Comparing to Other Benchmarks

In fig. C.19 we plot the performance on the individual IMAGENET-C corruptions for the models trained on clean and obfuscated images from fig. 12. We see that the gains from training on obfuscations become bigger for the higher corruption strength (each IMAGENET-C corruption has 5 different strength levels) but that this seems to be limited to a few of the corruptions, mainly the ones that apply different kinds of noise to the image. This might be because some of our obfuscations force the models to classify images with reduced information similar to adding noise. That these strong performance gains are limited to the simpler corruptions further shows that our obfuscations are very different from the IMAGENET-C corruptions. Similar to what we saw in fig. C.13 it might also be the case that the generalization we can see is not reversible, i.e. that models that are trained with data augmented by the noise corruptions will not become significantly more robust to our obfuscations.

## C.7    Top-K Accuracy

While the obfuscations drastically reduce the accuracy of the models it might be possible to counteract this by being more permissible in assigning a class label to an image if it is in the top-$k$ predicted (super-)classes. We show results for top-1 (same as fig. 4), top-3 and top-5 accuracy for the best model out of each pretrained category in fig. C.20. This leads to a significant increase in accuracy, up to $87.6\%$ worst case accuracy for the CLIP ViT L14 model. This is still a significant reduction from the almost perfect accuracy that the models now achieve on clean images. Additionally using top-5 accuracy for 16 super-classes will lead to a significant increase of false positives if we e.g. assume these classes were different types of policy violations. Therefore using top-k accuracy for $k > 1$ will only be a viable defense in cases where over-triggering of the classifier is not a concern. An interesting observation is that the ordering of the models changes in some cases, e.g. the CLIP ViT L14 model is the strongest model for top-3 and top-5 accuracy, while the ViT-B8 is stronger for top-1.

## C.8    Results Without Class Reweighting

As described in section 3.3 we weight the accuracy per class by the inverse number of image per class as to not have the biggest classes dominate the overall accuracy. In fig. C.21 we show the results for the best pretrained models of each category without reweighting. If you compare the accuracies with the class reweighted ones in the top of fig. C.20 you see there are clear differences, e.g. the worst-case accuracy of the best CLIP model drops from 55.5% to just 28.8%, most likely because the clip model does not have good performance on one of the biggest classes. In general the results seem more noisy and extreme without reweighting and there is no clear trend of increase or decreased performance across either obfuscations or models. We can see this as well when looking at the standard deviation of the top 1 accuracy for 5 different random seeds with and without reweighting in fig. C.22.

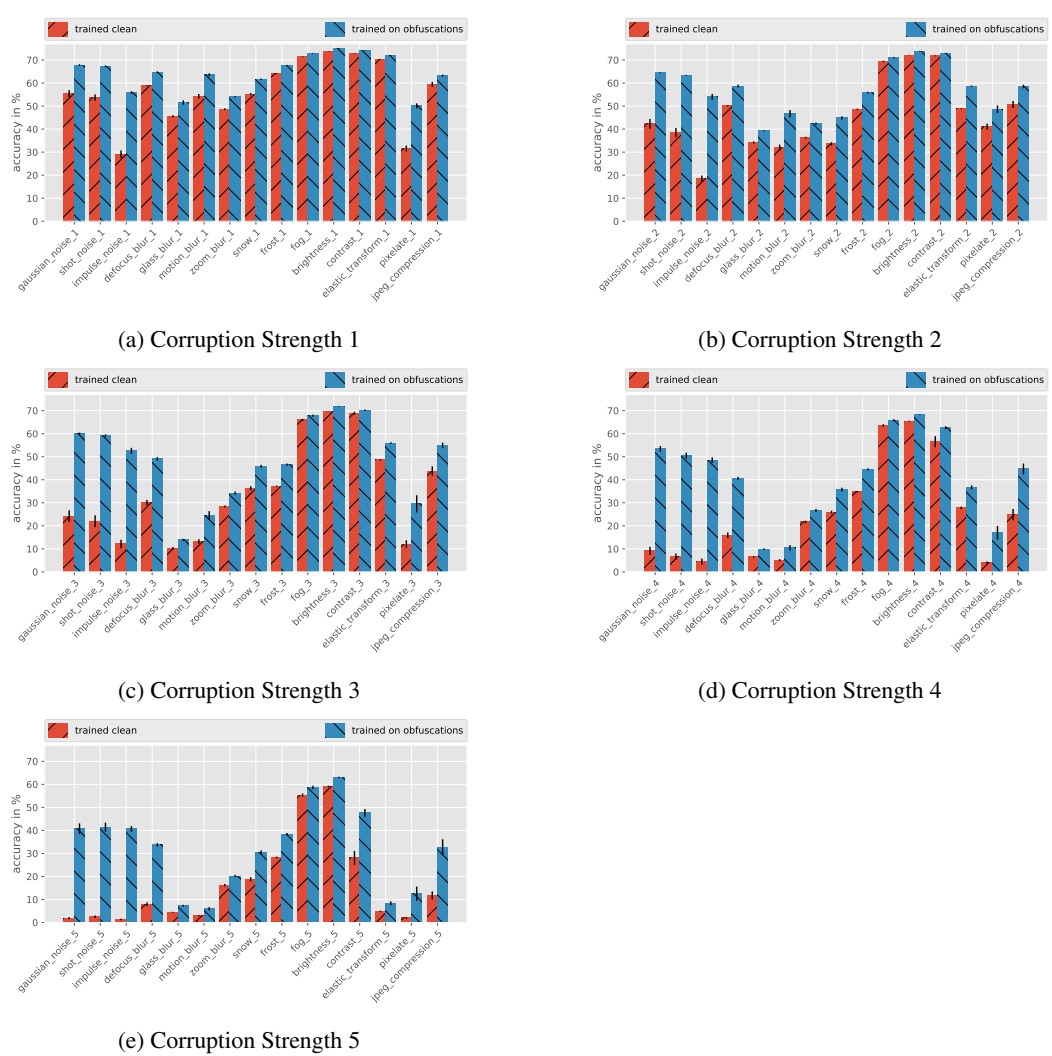

(a) Corruption Strength 1

(b) Corruption Strength 2

(c) Corruption Strength 3

(d) Corruption Strength 4

(e) Corruption Strength 5

Figure C.19: Comparison of top 1 accuracy when training on only clean and training on the training obfuscations for individual IMAGENET-C corruptions.

## C.9 Class Confusion

To investigate the mistakes models make through obfuscations we look at class confusion matrices for the ResNet50 v2 taken from Tensorflow Hub in section 4.1. figs. C.23 to C.26 show the percentage of images of the true class that were classified as the predicted class for clean images and all training and hold-out obfuscations. While there are some confusions between semantically similar classes like *Dog* and *Bear* or *Car* and *Van / Truck* that are simply amplified by the obfuscations we can see some interesting patterns. Some obfuscations lead to general misclassifications, i.e. classifying a large percentage of images as a specific class. Examples of this are *PerspectiveTransform* leading to many images being classified as *Clock* or *Texturize* leading to many images being classified as *Elephant*. The former might come from the fact the transformation on the black background makes the original image look like a clock on a wall, while the latter most likely comes from the fact that some of the textures are grey like elephant skin.

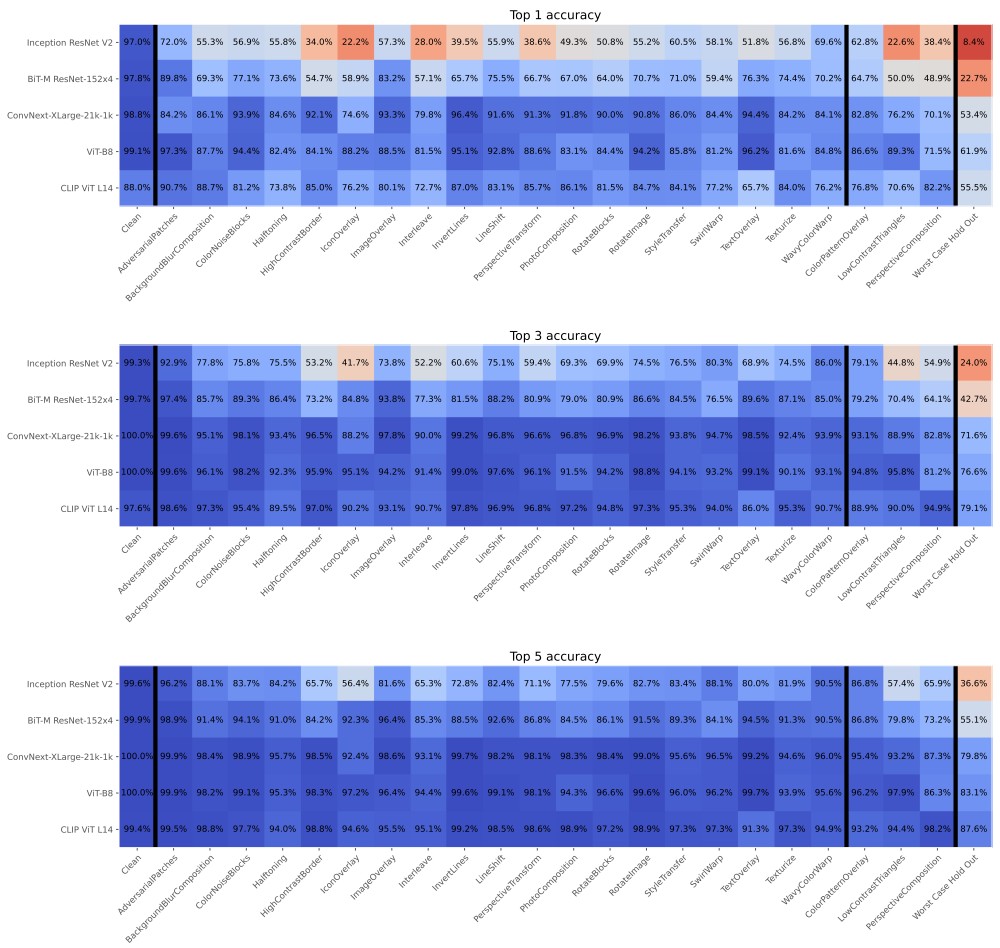

Figure C.20: Top 1 (top), top 3 (middle) and top 5 (bottom) accuracy for the best pretrained models of each category.

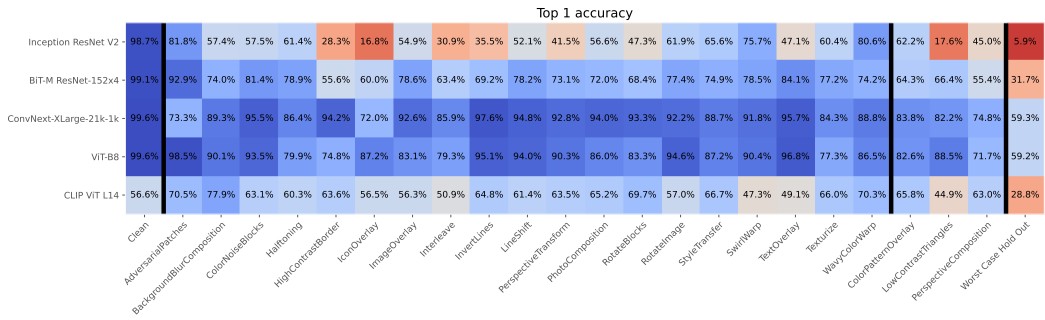

Figure C.21: Top 1 accuracy for the best pretrained models from each category without class reweighting.

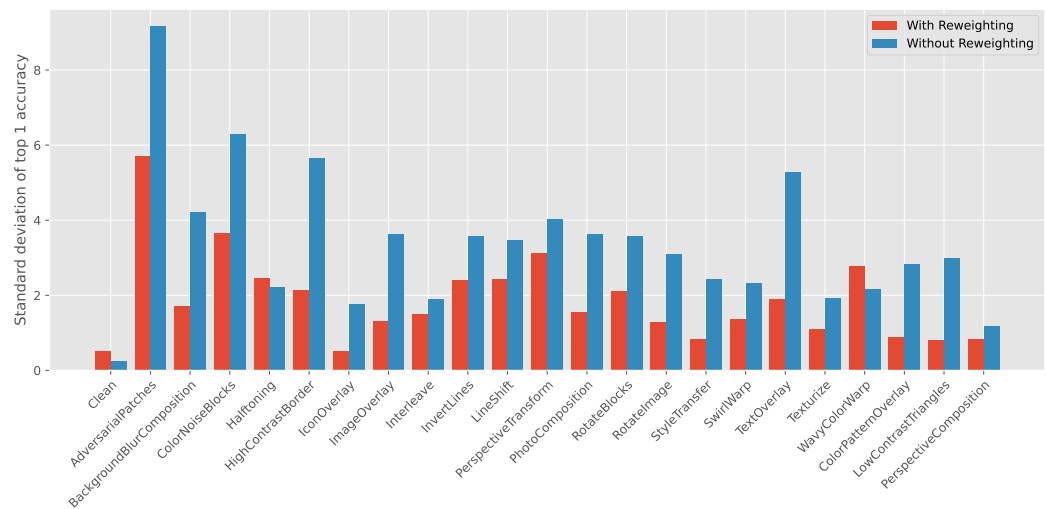

Figure C.22: Standard Deviation of top 1 accuracy for 5 ResNet50 models with different random seeds trained only on clean images.

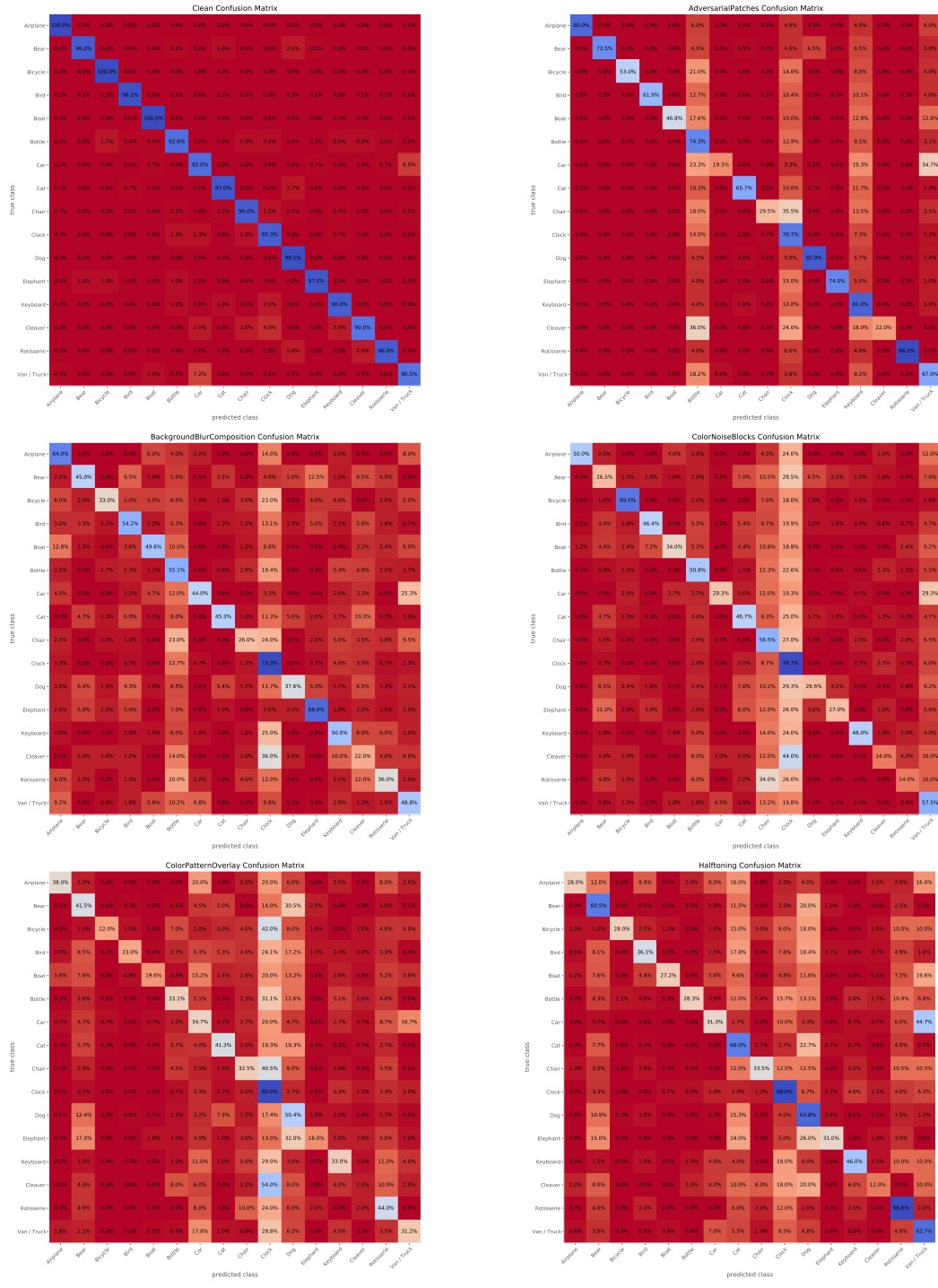

Figure C.23: Class confusion matrices for the different obfuscations

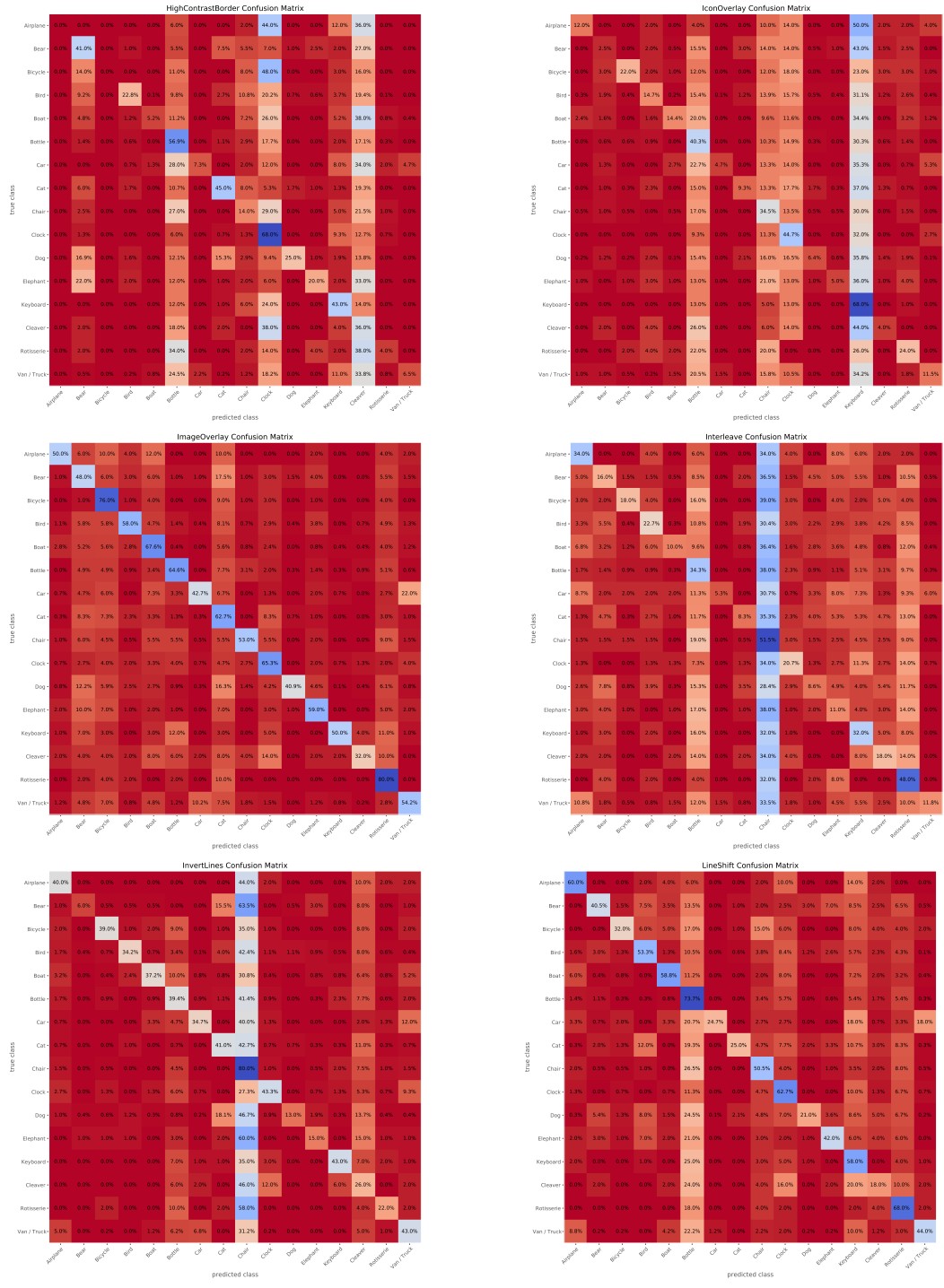

Figure C.24: Class confusion matrices for the different obfuscations

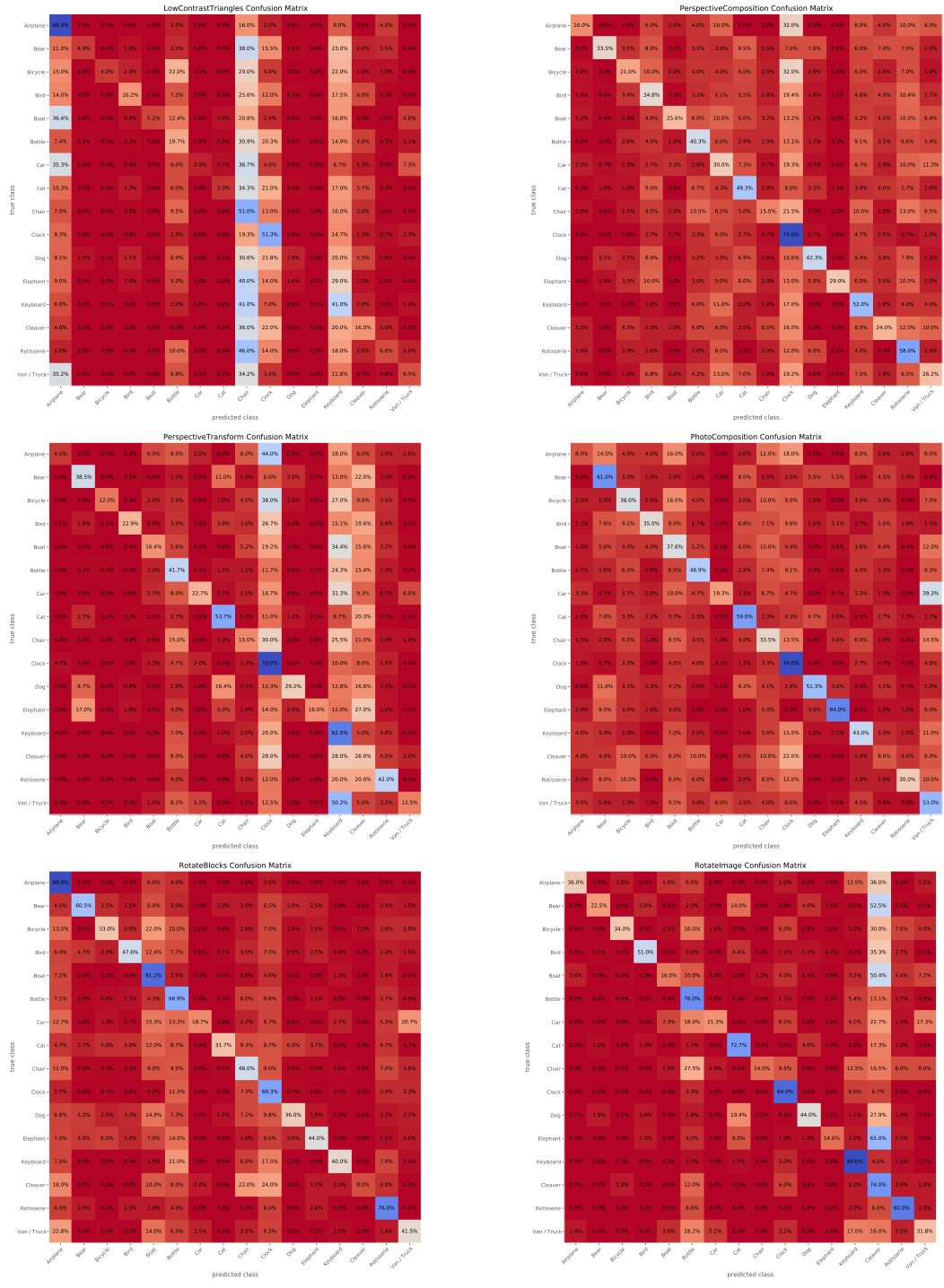

Figure C.25: Class confusion matrices for the different obfuscations

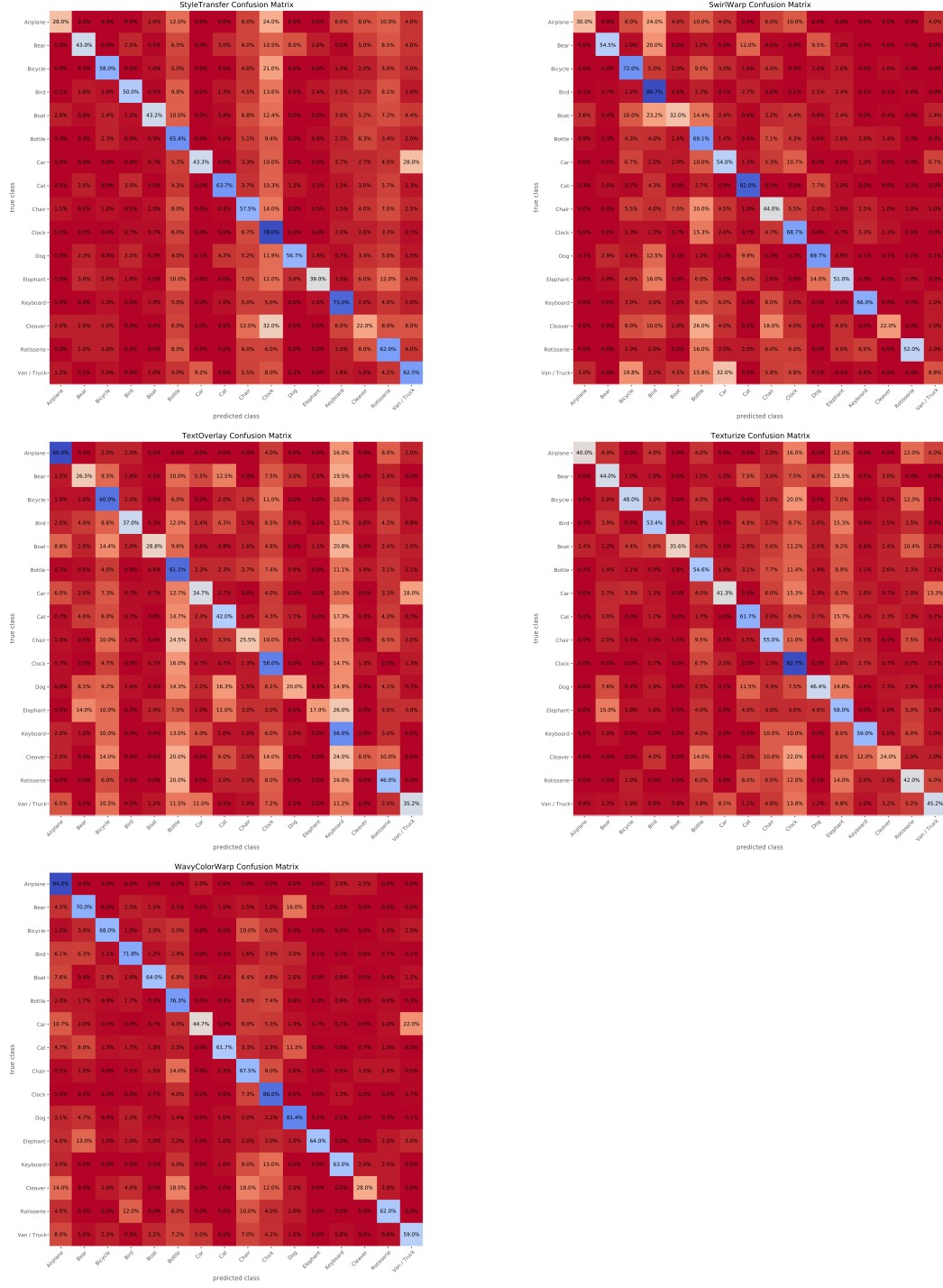

Figure C.26: Class confusion matrices for the different obfuscations

