# OpenReview forum: "Benchmarking Robustness to Adversarial Image Obfuscations"
_NeurIPS.cc/2023/Track/Datasets_and_Benchmarks — NeurIPS 2023 Datasets and Benchmarks Poster_

### Official Review · Reviewer_nVKg · 2023-07-18
**Dataset and evaluation of adversarial image obfuscations**

**Rating:** 7
**Confidence:** 5

**Strengths:**

- The dataset aims to benchmark a real-life issue (robustness to image obfuscations that aim to bypass safety systems) and is thus of significant interest to the NeurIPS community
- The dataset design is inspired by real-life attacks that the authors experience of Google systems
- The authors additionally provide thorough benchmarks of various pretrained ImageNet models, perform analyses of different augmentation/OOD techniques, adversarial training, and show interactions between the subsets.
- The authors demonstrate that the explored problem is not yet solved (as shown by the low test accuracy of tested approaches) and remains an active research field in which this dataset can be used for benchmarking.
- I deeply value and respect the authors' choice to be upfront about their prior CVPR23 rejection and a summary of changes. It looks to me as if all "issues" were resolved.
- Datasheet included
- Public repo for dataset

**Additional Feedback:**

- Are there any plans to support a public leaderboard ala RobustBench? I think this would help to propagate the dataset but also help researchers working on this problem to better identify current SOTA approaches.

**Clarity:**

The paper is clearly written and easy to follow. Super minor: There are a couple of missing spaces after eg.. and wrong "s".

**Correctness:**

The claims made in the submission are correct. However, I am not sure about the AT results (see above).

**Documentation:**

There is sufficient detail on data collection and organization, availability and maintenance, and ethical and responsible use. However, I would propose to add a "download all" link.
The benchmarks are reproducible except AT.

**Ethics:**

No (not addressed) ethical concerns.

**Limitations:**

Authors adequately addressed the limitations and potential negative societal impact of their work

**Opportunities For Improvement:**

- I have some concerns about the adversarial training (AT) results. 1) The authors state the used "2-step FGSM-based PGD"-training. That makes no sense. Is it FGSM or PGD? 2) I cannot find any info about the mitigation of robust overfitting (Rice et al. "Overfitting in adversarially robust deep learning"). Do the authors use early stopping or similar?
- To ensure that the obfuscations do not introduce a new semantic meaning the authors manually inspect 50 examples per transformation and allow a 4% error for the selected obfuscation hyperparameter. 1) the sample size is quite low. It would be better to inspect a larger fraction 2) it seems like the authors inspect themselves. It would be more appropriate to use an independent 3rd party.

Minor:
- As the authors mention themselves, one limitation is that fixed datasets can be in principle overfitted (even if they are as large as this one).
-  L143ff "we make sure that the chosen images do not introduce any objects which could be categorized into one of the 16 super-classes": how do the authors achieve this?
- L306: "we decided against releasing the code to create the obfuscations systematically". I understand the concern, but most of these obfuscations should be easily to reproduce just by looking at the images but the authors additionally state details in the appendix.

**Relation To Prior Work:**

The authors discuss relations to previous ImageNet-based datasets as well as adversarial attacks. One little addition I would like to propose is to add that Geirhos not only introduced the cue-conflict dataset but also multiple corruptions in the paper that the authors cited.

**Summary And Contributions:**

The authors propose a dataset of 22 transformations of ImageNet to benchmark the robustness to adversarial image obfuscations. The additionally evaluate 33 pretrained ImageNet models and show that scaling data/parameters helps to generalize. Additionally, they evaluate multiple augmentation techniques and show that MixUp is usually the most effective mode. Further, they show that distribution shift algorithms and adversarial training do not improve the performance. Neither does training on (a different set) of the proposed augmentations.

---

> ### Author Response · Authors · 2023-08-18
>
> Thank you for the review!
>
> Regarding adversarial training, you are right that our formulation was confusing. The model is trained with 2-step PGD with a step size of ⅝ epsilon ~0.01. We changed the formulation to just mention PGD. We also added a reference to [1] as we used their parameters they used for strong ImageNet performance.
> We used early stopping to avoid robust overfitting.
>
> We agree that more samples and an independent process would be better but other datasets like ImageNet-C also have a significant number of images that become too corrupted to identify. Even the original ImageNet has images that are hard to classify uncorrupted. In section C.2 we tried to estimate an upper ceiling for the worst-case accuracy which implies that there is still a large gap above the best model’s performance.
>
> The images that are used in some obfuscations were chosen manually and because we have just 16 superclasses we could ensure they don’t include any content that should be categorized into any of the superclasses.
>
> Regarding our release process. While we agree that the obfuscations could be replicated, we made this choice to balance the reproducibility with not making it too much easier for attackers, especially ones which would use premade tools but not make the effort to implement something themselves.
>
> Thank you for the remarks about minor typos, we have fixed them in the new version.
>
> We added a mention of the corruptions that Geirhos introduced to the related work section.
>
> We are happy to add a public leaderboard on the github page if other researchers use our benchmark to evaluate other models.
>
> [1] S.-A. Rebuffi, F. Croce, and S. Gowal, “Revisiting adapters with adversarial training”. Proceedings of the Eleventh International Conference on Learning Representations, 2023

---

> > ### Comment · Reviewer_nVKg · 2023-08-22
> > **Reviewer Response**
> >
> > 1. Regarding label integrity of corruption:
> > I agree that ImageNet-C has unrecognizable instances, but the problem definition is fundamentally different. Hendrycks attempts to benchmark robustness between models/training techniques and then it's acceptable if some samples are not recognizable because they are unrecognizable to all models (and the benchmark averages over multiple severities). While ImageNet-C is not perfect, it is probably fine for that case.
> > Your dataset, however, is motivated by a comparison against human attackers. There are no gains for malicious actors in attacking safety systems with unrecognizable images. E.g., you cite pornography spam. If the corruption breaks the image to the point where there's "nothing left to see" then it's actually fine to accept this image. In this case, your dataset may overestimate the risk.
> > And I am not saying this to reject your paper, I still think it's good - but this is a limitation that you should be open about.
> >
> > 2. Regarding your comment to reviewer Y2nL: Yes, please implement this in PyTorch!
> > 3. This is not important to the rebuttal but it would be helpful to provide some option to download the dataset at once (either a script or one link). From my own experience with ImageNet-C, I am so annoyed that I have to download the corruption categories one by one.
> >
> > All other issues are resolved. Thanks and good luck with your publication/dataset! I am firm on "7: Accept".

---

> > > ### Author Response · Authors · 2023-08-30
> > >
> > > Thank you again for the response!
> > >
> > > 1. We add to the limitations that there are images that are too corrupted and readers should take that into account when interpreting our metric.
> > > 2. We will try to add code to evaluate PyTorch models before the conference.
> > > 3. We will add scripts to download all the subsets together.

---

### Official Review · Reviewer_jWzW · 2023-07-21
**Benchmarking Robustness to Adversarial Image Obfuscations**

**Rating:** 7
**Confidence:** 4
**Correctness:** The reported results appear to be cor…
**Clarity:** The paper has clarity in its experime…

**Strengths:**

S1. They propose an exhaustive set of obfuscations grouped into 5 categories.
   S2.  They conduct a methodical set of experiments that bring out the intuitions quite succinctly.
   S3.  The flow and writing of the manuscript is clean and thorough.

**Additional Feedback:**

NA

**Documentation:**

Very good.

**Limitations:**

It’s appreciative that the paper has a section on limitations where it poses some of the limitations of the work and the possible future works for the research community. For the rest, refer to Opportunity for Improvement.

**Opportunities For Improvement:**

Below I mention some of the missed opportunities or queries that I have.

   I1.  Since ImageNet-C has natural degradations and considering the degree of the proposed obfuscations, it would be good to compare the performance of the models on some of the important corruptions of ImageNet-C (like impulse noise or glass blur) to observe the difference in performance with the held out obfuscations proposed in this work. This may help in understanding the adversarial strength that the proposed obfuscations bring in.
    I2.  Would it be relevant to observe the model performance also with the different renditions of the classes, that are proposed in ImageNet-R[1]?
    I3.   The computation of the ‘worst-case’ accuracy that has been reported along with the held out obfuscations isn’t very clear from the main paper. It would be better to have the definition of the metrics apriori in the main paper. Appendix A.8 defines the steps, where weighing the average accuracy with the inverse of the super class occurrence in step 7 is not very intuitive.

[1] : Hendrycks, Dan, et al. "The many faces of robustness: A critical analysis of out-of-distribution generalization." Proceedings of the IEEE/CVF International Conference on Computer Vision. 2021.

**Relation To Prior Work:**

The authors have described the related works relevant to their piece of work appropriately.

**Summary And Contributions:**

This paper focuses on proposing a set of transforms for the ImageNet dataset, that is different from the existing ImageNet-C dataset which consist of the simple transformations like blur, rotate, shear and more, and also holds a difference from the adversarial robust methods which perturbs images making them imperceptible to humans. This work aims at enabling research to make visual classifiers robust to adversarial obfuscations that can be able to fool the classifiers but comprehendable to humans. Their contribution essentially lies in curating a set of 22 strong compositional obfuscations and measuring generalisation on unknown obfuscations using 33 pre-trained models.

---

> ### Author Response · Authors · 2023-08-18
>
> Thank you for the review!
>
> 1. Figure C.19 in the appendix has individual results on all the ImageNet-C corruptions for a model trained only on clean and a model trained on the training obfuscations. Section C.6 discusses these results and shows that training on our obfuscations increases a model’s robustness to noise corruptions, including impulse noise, significantly. The effect on other corruptions, including glass blur, is a lot weaker. Similarly we can see in figure C.18 that training on the training obfuscations only has a large effect on one of the hold-out obfuscations (ColorPatternOverlay) while the accuracy on the other is only affected marginally.
> 2. We haven’t evaluated on the 200 classes that ImageNet-R uses and it might extend the discussion too much and is a lot higher than what you would normally encounter in a content filtering setting. We agree that this could be an interesting direction for the future as more classes could allow a more fine-grained analysis, especially of stronger models.
> 3. We moved the definition of the metric from the appendix back to the end of section 3.3. The reason why we weigh the accuracy by the inverse occurrence is that some super-classes are very overrepresented (e.g. dogs make up slightly more than half the images) and we want to avoid the end result being dominated too much by single super-classes. We added this justification in section 3.3 and added section C.8 to the appendix which compares the metric with and without this weighting for a few models showing that weighting reduces the noise in the accuracy metric.

---

> > ### Author Response · Authors · 2023-08-30
> >
> > Dear Reviewer jWzW,
> >
> > As the discussion window is closing soon we would like to ask if our response addressed your concerns or if there are any further questions.

---

> > > ### Comment · Reviewer_jWzW · 2023-08-30
> > > **Reviewer response**
> > >
> > > Thank you for the responses, I have also increased the score of my prior scores. All the best for your paper!

---

### Official Review · Reviewer_Y2nL · 2023-07-22
**Review for "Benchmarking Robustness to Adversarial Image Obfuscations"**

**Rating:** 5
**Confidence:** 3
**Correctness:** The dataset and benchmark appear to b…

**Strengths:**

1. The paper aims to address an important challenge in automated content filtering and moderation, which is crucial for building thriving user communities and preventing abuse.
2. The paper introduces a new benchmark to measure the performance of classifiers on obfuscated images.
3. The paper presents comprehensive evaluations.


**Additional Feedback:**

It would be better if the code could improve interoperability across deep learning frameworks.

**Clarity:**

The paper is clear and easy to follow.


**Documentation:**

It is reasonably well documented.

**Ethics:**

In section 5.2, the authors opted to release a precomputed dataset instead of open-sourcing the obfuscation algorithms. This approach may provoke debates. It benefits the community greater to have security by design rather than security by obscurity.

**Limitations:**

Limitations have been adequately discussed.

**Opportunities For Improvement:**

1. The authors correctly acknowledge that $\ell_p$ adversarial threat models are inadequate in safeguarding online platforms. However, this paper lacks considerations on white-box and black-box transferability attacks beyond $\ell_p$ perturbation budgets, which are stronger than the static obfuscation approach examined in this paper. Adversarial training beyond the $\ell_p$ budgets could also have additional benefits towards unforeseen attacks [a].
2. While the reviewer appreciates the variety of obfuscation methods presented, the paper however did not justify the choices of methods and hyper-parameters. Attackers may easily bypass the defenses with new obfuscation strategies or different hyper-parameters. In that sense, it is difficult to convince the reader that the approach of exhaustively testing all previous methods could be effective for future threats.
3. The algorithms for data construction are not provided and instead a dataset is constructed and tuned for ImageNet. It is not clear how this dataset could be used to effectively increase robustness in the real world for the specific applications of other practitioners facing unforeseen attacks.
4. The experimental results are extensive, but lack proper detailed observations and related discussions.

[a] Laidlaw et al., Perceptual Adversarial Robustness: Defense Against Unseen Threat Models, ICLR 2021.

**Relation To Prior Work:**

It appears to be an extension to previous image corruption datasets / benchmarks. Section 2 introduces the related datasets but it would be better to include a statement to differentiate this work from them.

**Summary And Contributions:**

The paper introduces a new benchmark to measure the performance of classifiers on obfuscated images. The benchmark is based on ImageNet and simulates the type of obfuscations created by malicious actors. The paper presents a set of 22 strong, diverse, adversarial obfuscations to provide a concrete starting point to make and measure progress in obfuscation robustness.

---

> ### Author Response · Authors · 2023-08-18
>
> Thank you for the reference to [a] — we have included it in our discussion of methods that may be adapted to the case of obfuscations (rather than perturbations) to generalize beyond obfuscations seen in a training set. However, note that [a]’s method also falls in the case of methods seeking to defend against attacks that are ‘imperceptible’ to human observers. The major focus of our work deals with attack scenarios where manipulated images can still be “harmful” even when human observers can clearly make out that they have been adversarially modified as long as the underlying semantic content is decipherable.
>
> Note that our benchmark does measure the ability of training/defense strategies to generalize to unseen threat models by having a subset of obfuscations that are held out during training.
>
> Also, we do not delve too deeply into white-box attacks since in the attack scenarios we consider against online platforms, attackers don’t have access to the classification models.
>
> Section 3.2 (from line 123 on) talks briefly about how we chose the hyperparameters. Obviously the choice of the obfuscations is somewhat arbitrary. We both incorporated obfuscations that are similar to what we have seen being used by attackers as well as coming up with novel methods that fool most systems. As we wrote in the paper, this list can never be exhaustive but we hope that our setup allows to find methods that can generalize better than previous models towards unseen obfuscations.
>
> After discussion with our ethics & safety committee, we decided to not release the obfuscation code itself, since it may make it easier for attackers to target the system vulnerabilities pointed out in our paper. While this does not allow practitioners to use our obfuscations to directly improve their systems, this was not the intention of our benchmark. Instead we hope that our benchmark can help find methods that can leverage obfuscations (ours or from other sources) better or can even increase robustness without the need of specific obfuscations, e.g. a general augmentation approach.
>
> Are there specific experimental results that you would like to have more discussion on? We would be happy to expand where needed.
>
> We don’t think our benchmark is an extension of previous ones as the setup, metric and obfuscations are all different. Previous benchmarks either featured natural OOD samples, natural distribution shift or tried to emulate natural corruptions or they were focused on perceptually minimal transformations that often required access to the model. Our benchmark includes strong obfuscations, that are allowed to change the image drastically as long as the class label stays intact. The obfuscations do not need model access and we also structure our benchmark differently to other popular examples, e.g. ImageNet-C, by allowing training on a subset of obfuscations. We have extended section 2 to make the distinction to the different types of existing benchmarks clearer.
>
> If there is demand we will implement ways to evaluate models in pytorch.

---

> > ### Author Response · Authors · 2023-08-30
> >
> > Dear Reviewer Y2nL,
> >
> > As the discussion window is closing soon we would like to ask if our response addressed your concerns or if there are any further questions.

---

### Official Review · Reviewer_hqY3 · 2023-07-22
**Benchmarking Robustness to Adversarial Image Obfuscations**

**Rating:** 8
**Confidence:** 4
**Clarity:** Yes

**Strengths:**

The paper has designed a new benchmark for curating obfuscations approach in image attack. The paper had created, curated and tuned a set of 22 strong, diverse, adversarial obfuscations.  The authors evaluated 33 different pretrained models on our benchmark and trained models on over 60 subsets of our training obfuscations. Hence, the experimental part is very strong. The proposed method is general and broad.

**Additional Feedback:**

None

**Correctness:**

For this benchmark paper, the evaluation methods and experiment seem appropriate and performed correctly.

**Documentation:**

Yes

**Ethics:**

Yes

**Limitations:**

The authors seem didn't discuss the limitations and potential negative societal impact of their work. Please add this part in the final version.

**Opportunities For Improvement:**

There are some minor issues:

1. Please polish the paper thoroughly. For example, in the Abstract, "It goes beyond IMAGENET-C and IMAGENET-C¯ by proposing ...", the  IMAGENET-C had been repeated twice here.

2. It is better not to add too many references at one place. For example, in page 1,  "considerable research [1–5], with multiple benchmarks [6–10] “。 Please try to illustrate one by one for the references.

3. It is better to illustrate the practical potentials of the proposed design.

**Relation To Prior Work:**

Yes

**Summary And Contributions:**

In this paper, the authors aim to tackle a more common adversarial threat than the one considered by lp-norm bounded adversaries. The paper proposed some general, drastic, adversarial modifications that preserve the original content intent that goes beyond IMAGENET-C. The paper claims that this benchmark is the first one that curates obfuscations similar to what bad actors use to circumvent content filter models.

The paper is well written. However, there are some minor issues:

1. Please polish the paper thoroughly. For example, in the Abstract, "It goes beyond IMAGENET-C and IMAGENET-C¯ by proposing ...", the  IMAGENET-C had been repeated twice here.

2. It is better not to add too many references at one place. For example, in page 1,  "considerable research [1–5], with multiple benchmarks [6–10] “。 Please try to illustrate one by one for the references.

3. It is better to illustrate the practical potentials of the proposed design.

---

> ### Author Response · Authors · 2023-08-18
>
> Thank you for your review!
>
> ImageNet-C̄ is actually a different dataset based on ImageNet-C from the paper titled “On Interaction Between Augmentations and Corruptions in Natural Corruption Robustness”. It is referenced as [11] in the new paper (it was [25] in the old version) and we mention it in the related work section.
>
> We changed parts of the introduction to not cite too many references together and distinguish them better —see the updated paper pdf.
>
> We have added an extended discussion about the practical potential at the end of the introduction, as suggested. We are happy to incorporate more if you have suggestions for specific directions.
>
> Regarding limitations and potential negative societal impact, we address these in section 5.1 and 5.2 respectively. Please let us know if there is anything additional that you would want us to discuss.

---

> > ### Author Response · Authors · 2023-08-30
> >
> > Dear Reviewer hqY3,
> >
> > As the discussion window is closing soon we would like to ask if our response addressed your concerns or if there are any further questions.

---

### Official Review · Reviewer_W7ys · 2023-07-22
**A significant contribution for the security of real-world ML systems**

**Rating:** 8
**Confidence:** 4
**Clarity:** The paper is very well-written and ea…

**Strengths:**

- The paper is very well motivated, as the usage of this type of obfuscations by malicious users is extremely realistic for systems deployed to the real-world.
- The paper provides extensive experiments showing how models are poorly robust to these perturbations and how training on a set of obfuscations makes the model robust to a held-out set. They aso observe that adversarial training and distribution shift algorithms do not help.

**Additional Feedback:**

N/A

**Correctness:**

The claims are correct, and the evaluation methods and experiment design are designed and performed correctly.

**Documentation:**

The authors provide extensive documentation on how to retrieve the dataset, and provide well-documented code to automatically evaluate TensorFlow models using this dataset.

**Ethics:**

The authors acknowledge that such dataset could be used for malicious purposes by adversaries. For this reason, they release only the images with the applied perturbations, without releasing the tool to generate them. In this way, they ensure that researchers can benchmark their models/training techniques against these perturbations, without letting malicious users leverage it for unethical purposes. Hence, I believe that no ethics review should be required.

**Limitations:**

The authors discuss a series of limitations which, however, do not affect the relevance of this work.

**Opportunities For Improvement:**

- The authors do not discuss how training on these obfuscations affects the accuracy on clean images. Is there a trade-off? What does it look like?
- A possible defense against these attacks could be checking whether a given class is among the top-k classes predicted by the model (i.e., if any NSFW class appears in among the top-k classes, then the input should be flagged). For this reason, it could be interesting if the original class still appears in the top-k (e.g., 5) classes.

**Relation To Prior Work:**

The authors discuss how this work differs from previous contributions and benchmarks. However, they may want to consider mentioning AutoAttack [1] among the $\ell_p$ attacks as it is the one of the most evaluation methods used to assess robustness against these attacks.

Moreover, the authors may want to take a look at the work by Apruzzese et al. [2], who pointed out how attackers are more likely using obfuscations like those proposed by the authors, rather than gradient-based ones.

[1] Croce, Francesco, and Matthias Hein. "Reliable evaluation of adversarial robustness with an ensemble of diverse parameter-free attacks." International conference on machine learning. PMLR, 2020.

[2] Apruzzese, Giovanni, et al. "“Real Attackers Don't Compute Gradients”: Bridging the Gap Between Adversarial ML Research and Practice." 2023 IEEE Conference on Secure and Trustworthy Machine Learning (SaTML). IEEE, 2023.

**Summary And Contributions:**

The paper introduces a benchmark to evaluate the robustness of classification models to adversarial obfuscations. The authors propose a series of obfuscations that could be used by malicious users to evade classification models deployed to the real world, in a full black-box setting. They verify whether a set of models not trained on these perturbations is robust to them, and which models perform better, both in therms of architecture and size. Moreover, the authors study if and how training on a set of these perturbations works, and they interact with each other.

---

> ### Author Response · Authors · 2023-08-18
>
> Thank you for your review!
>
> Regarding the clean accuracy, figure C.14 in the appendix shows the results for models trained on an increasing number of obfuscations. The first row is a model only trained on clean data while the last row is trained on all 19 training obfuscations. The accuracy on clean images is in the first column and seems to even slightly increase by adding more obfuscations. We have added this observation to the end of section 4.3.1 in the revised paper.
>
> We added results for top-3 and top-5 accuracy in the appendix section C.7. The trends from the top-1 accuracy results hold mostly true here, but there are some ordering changes (e.g. the CLIP model being the strongest. There is also a trade-off here as using e.g. top-5 accuracy will produce a lot more false positives, especially since we only have 16 superclasses, which might make this defense not useful in a lot of cases.
>
> Thank you for the additional references. We agree that AutoAttack has played an important role in truthfully verifying accuracy to adversarial attacks and we added a reference to it. [2] is indeed a very relevant paper we weren’t familiar with. We have cited it in our introduction—see the updated paper pdf.

---

> > ### Comment · Reviewer_W7ys · 2023-08-19
> >
> > Thank you for your reply and for updating your paper. As my (small) concerns are now addressed, I am increasing the score.
> >
> > Just one very minor note: in figure C.20, the title of all three images is "Top 1 accuracy", while I my understanding from the caption is that it should be "Top {1,2,3} accuracy". Is it a typo?

---

> > > ### Author Response · Authors · 2023-08-21
> > >
> > > Thanks for updating your score. The titles were indeed a typo and we've updated them.

---

### Author Response · Authors · 2023-08-18

We thank all reviewers for their thoughtful comments and suggestions we have updated the pdf and posted answers to the reviews in the respective threads.

---

### Decision · Program_Chairs · 2023-09-22

**Decision:**

Accept (Poster)

**Comment:**

This paper provides a benchmark for image obfuscations carried out by attackers aiming to fool some image classifier/detector. The benchmark provides a dataset of images obfuscated using 22 different different obfuscation, with 19 used in training and 3 held-out. They evaluate the effectiveness of training with obfuscations across different architectures, and resultant performance on the held-out obfuscations. While there were some concerns around the generalizability of the obfuscations and the use of only $\ell_p$ attacks in the adversarial example setting, the reviewers largely agreed the paper represents a good new benchmark for training models robust to varied obfuscations in the future, along with good preliminary exploration of baselines.